# Isolation of Main Pathogens Causing Postharvest Disease in Fresh *Codonopsis pilosula* during Different Storage Stages and Ozone Control against Disease and Mycotoxin Accumulation

**DOI:** 10.3390/jof9020146

**Published:** 2023-01-21

**Authors:** Bingyu Lv, Xi Yang, Huali Xue, Mina Nan, Yuan Zhang, Zhiguang Liu, Yang Bi, Suqin Shang

**Affiliations:** 1College of Science, Gansu Agricultural University, Lanzhou 730070, China; 2College of Food Science and Engineering, Gansu Agricultural University, Lanzhou 730070, China; 3College of Plant Protection, Gansu Agricultural University, Lanzhou 730070, China

**Keywords:** *Codonopsis pilosula*, postharvest disease, morphological and molecular biology identification, ozone, mycotoxin accumulation

## Abstract

*Codonopsis pilosula* is an important Chinese herbal medicine. However, fresh *C. pilosula* is prone to decay during storage due to microorganism infections, seriously affecting the medicinal value and even causing mycotoxin accumulation. Therefore, it is necessary to study the pathogens present and develop efficient control strategies to mitigate their detrimental effects on the herbs during storage. In this study, fresh *C. pilosula* was collected from Min County in Gansu Province, China. The natural disease symptoms were observed during different storage stages, and the pathogens causing *C. pilosula* postharvest decay were isolated from the infected fresh *C. pilosula*. Morphological and molecular identification were performed, and pathogenicity was tested using Koch’s postulates. In addition, the control of ozone was examined against the isolates and mycotoxin accumulation. The results indicated that the naturally occurring symptom increased progressively with the extension of storage time. The mucor rot caused by *Mucor* was first observed on day 7, followed by root rot caused by *Fusarium* on day 14. Blue mold disease caused by *Penicillum expansum* was detected as the most serious postharvest disease on day 28. Pink rot disease caused by *Trichothecium roseum* was observed on day 56. Moreover, ozone treatment significantly decreased the development of postharvest disease and inhibited the accumulations of patulin, deoxynivalenol, 15-Acetyl-deoxynivalenol, and HT-2 toxin.

## 1. Introduction

*Codonopsis pilosula*, a perennial herb of the *Campanulaceae* family, is a traditional Chinese herb that is rich in bioactive compounds including polyacetylene, polyene, flavonoids, lignans, alkaloids, coumarins, terpenes, steroids, organic acids, and polysaccharides [1]. *C. pilosula* is widely used in the therapy of hyperlipidemia, asthma, bronchitis, tuberculosis, and dyspepsia [2], and also plays a key role in boosting the body’s immunity [3], reducing blood glucose [4], promoting hematopoiesis [5], protecting the cardiovascular system [6], repairing damaged nerve cells, and regulating gastrointestinal function [7].

Owing to its unique climate, Gansu Province in China has more than 2000 years of history in cultivating *C. pilosula*. With its high commercial value and profitability, the cultivation of *C. pilosula* is continuously expanding. However, some soil-borne fungi, bacteria, and nematodes have adapted to the local host plants and natural environment in the region, thus, resulting in serious diseases that severely affect the yield and quality of *C. pilosula*. Research to date has focused on field disease of *C. pilosula*. For instance, Zhao et al. [8] suggested that *Fusarium oxysporum* is the predominant pathogen causing root rot of *C. pilosula* in Dingxi County, Gansu Province. Using molecular technology, Yu et al. [9] isolated and identified five pathogens in China: *Puccinia Campanumoeae* Pat causing rust disease, *Helicobasidium mompa* Tanaka causing violet root rot, *Sphaerotheca Codonopisi (Golov.)* Z.Y. Zhao causing powdery mildew, *Fusarium oxysporum* Schl. causing root rot, and *Septoria codonopsidis* Ziling causing blight of *Codonopsis* tangshen in Chongqing. Based on morphology and molecular characteristics, Wang et al. [10] isolated and identified *Septoria codonopsidis Ziling* causing leaf spot of *C. pilosula* in Gansu Province, while Chen et al. [11] proposed that *Botrytis cinerea* is the typical fungus causing gray mold in *C. pilosula*.

Modern research indicates that fresh Chinese herbs have higher active ingredients and pharmacological activities than those of dry products [12]. For instance, the contents of flavonoids, saponins, and polysaccharides in fresh *Astragalus* were 1.5 times more than those in dry products [13]. Because of this, the market for freshly harvested Chinese herbs has grown rapidly and is today more significant and well known than before. Unfortunately, the postharvest losses due to the disease of fresh Chinese herbs are quite severe. Nevertheless, no reports are available regarding the prevalence of pathogens during this period. In general, the herb is usually harvested in late autumn and allowed to dry naturally on the field for more than two months before being transported to the traders. The freshly harvested *C. pilosula* must be sufficiently dried (about 16% of water content), then stored for 3–6 years. If *C. pilosula* does not reach an optimum level of dry matter, its abundance storage of fat, starch, protein, sugar, and other organic substances may favor the development of some latent pathogens and ultimately result in serious postharvest disease. Fungal infections are thought to result in yearly losses of 15% to 25% [14]. Such postharvest losses cause enormous economic damage to *C. pilosula* processing industries, and more importantly, the product may completely lose its medicinal value, becoming contaminated with mycotoxins that have carcinogenic, teratogenic, and mutagenic toxicity [15]. It is therefore important to systematically study the postharvest diseases of freshly harvested *C. pilosula*, identify the pathogens present at various storage stages, and develop an efficient control strategy to mitigate their effects.

Ozone, an antioxidant compound, has been widely applied to manage the postharvest decay of fruits and vegetables. The efficacy of ozone in controlling postharvest disease is mainly ascribed to its strong inhibitory activity on pathogenic fungi. However, there are no reports on the influence of ozone on fresh Chinese herb medicine postharvest disease and mycotoxin production.

In this study, we collected fresh *C. pilosula* from Min County in Gansu Province, China, investigated the disease development of freshly harvested *C. pilosula* during storage, then isolated and identified pathogens causing *C. pilosula* postharvest disease based on morphological and molecular biological techniques during different storage stages. Finally, we examined the influence of ozone on fresh *C. pilosula* postharvest disease and mycotoxin production.

## 2. Materials and Methods

### 2.1. Sample

Samples of freshly harvested *C. pilosula* (cv. *Mindang*) were obtained from Min County (location 35° N and 104° E) in Gansu Province, China. Sample roots of similar size and without obvious pest or mechanical damage were selected in October 2020, and transported to the Chemical Biology Laboratory, College of Science, Gansu Agricultural University within 24 h after bagging, and stored at room temperature for further analysis.

### 2.2. Disease Development of Freshly Harvested C. pilosula during Different Storage Stages

Freshly harvested *C. pilosula* samples (without any processing treatment such as washing and drying) were placed directly in plastic bags (20 °C, 50% RH) in darkness for 7, 14, 21, 28, 42, and 56 days. Subsequently, the naturally occurring symptoms were observed and described. Different pathogens cause different disease symptoms, for example, in the early stages of root rot, small brown spots appear on the surface of the lower fibrous or lateral roots, with mild decay. As the disease expands, it gradually spreads to the main roots. The roots gradually decay from the bottom upwards and become dark brown and waterlogged [9]. Three replicates were included in each treatment, each treatment consisted of 50 samples of *C. pilosulas*, with a total of 900 samples being included in the whole experiment (50 samples × 3 replicates × 6 time points).

### 2.3. Isolation and Purification of Pathogens Causing Disease of Freshly Harvested C. pilosula during Different Storage Stages

Pathogens were isolated and purified from *C. pilosula* based on typical decay symptoms (the appearance of mycelium and spores on the surface of *C. pilosula*) [16]. Fragments (5 mm × 5 mm) showing typical disease symptoms during different storage stages were excised with sterilized scalpels from the junction of the healthy and diseased tissue, and the fragments were disinfected using 1% NaClO for 3 min, and then rinsed with sterile water three times to remove any NaClO residue. The treated fragments were placed onto potato dextrose agar (PDA) medium and cultured in darkness at 25 °C for 5 to 7 d. Single colonies were picked and transferred to a new PDA medium with a puncher. After 4 to 5 cycles of purification, a single purified colony was obtained.

### 2.4. Identification of Pathogens Causing Disease in Freshly Harvested C. pilosula during Different Storage Stages

#### 2.4.1. Morphological Identification

The preliminary identification was conducted based on colony morphology and macro and microconidia characteristics [17]. The spore suspension (1 × 10^6^ spores/mL) was inoculated on a PDA plate, and after drying, a cover glass was inserted into the PDA medium at an angle of 45°, and then cultured at 25 °C for 2 to 5 d. Subsequently, to observe colony morphology and pigment production, a 2 μL spore suspension was inoculated centrally in the PDA medium at 25 °C for 7 to 9 d. Colony morphology and pigment production were recorded on day 9. Mycelia were transferred using the copper pick-in method [18]. All samples were sprayed gold with an ion sputtering apparatus (MSP-1S, Shenzhen Research Precision Instrument Co., Ltd., Shenzhen, China) under 220 V and 40 mA, then the morphology of the spores was observed using a scanning electron microscope (SEM) (JSM-5910LV, Japanese electronics company, Tokyo, Japan).

#### 2.4.2. Molecular Identification

The morphologically identified pathogens were further subjected to molecular confirmation according to a previously used method [19] with some modifications. The pathogens were grown on PDA medium for 3 to 9 d, then the DNA of pathogenic mycelia was extracted using the CTAB method [20], and PCR amplification was performed using the primers *ITS1* (5′-TCCGTAGGTGAACCTGCGG-3′) and *ITS4* (5′-TCCTCCGCTTATTGATATGC-3′) for the 9 isolates. For isolates 21-1, 28-1, 28-2, and 56-1, the primers *Bt2a* (5′-GGTAACCAAATCGGTGCTGCTTTC-3′) and *Bt2b* (5′-ACCCTCAGTGTAGTGACCCTTGGC-3′) were employed for further PCR amplification. For the *Fusarium* strains (14-1, 14-2, 14-3), the special primers of *EF1*(5′-ATGGGTAAGGA(A/G)GACAAGAC-3) and *EF2*(5′-GGA(G/A)GTACCAGT(G/C)ATCATGTT-3′) were adopted [21,22]. Subsequently, the amplified products were detected using 2% agarose gel electrophoresis and clear bands were obtained.

The PCR amplification procedure was as follows: pre-denaturation at 94 °C for 5 min, denaturation at 94 °C for 10 s, annealing at 53 °C for 10 s, extension at 72 °C for 30 s, 3 cycles, and holding at 72 °C for 5 min. The amplified products were subject to electrophoresis with 2% agarose gel. The amplified fragments were sequenced by Beijing Bomede Biotechnology Co., Ltd., and the sequences were aligned to the NCBI (https://www.ncbi.nlm.nih.gov/ (accessed on 8 July 2021)) using BLAST for homology analysis. A phylogenetic tree was constructed by MEGA7.0 software (Molecular Evolutionary Genomics Analysis Version 7) using the neighbor-joining method.

### 2.5. Pathogenicity Test

The healthy and freshly harvested *C. pilosula* was thoroughly washed with tap water to remove any adhering soil and air dried. The herbs were surface disinfected with 0.1% NaClO for 15 min, rinsed with sterile water three times to remove excessive NaClO, and air dried again. Spore suspensions (1 × 10^6^ spores/mL) of the above isolated pathogens were prepared [18] and artificially inoculated by spraying onto the surface of the fresh and healthy *C. pilosula*. The control group was inoculated with sterile water. After natural drying, the inoculated *C. pilosula* were kept in darkness (20 °C, 50% RH). After an incubation period of 28 days, the disease symptoms were recorded, and different pathogens led to different disease symptoms [23]. The pathogens were re-isolated again from the *C. pilosula*’s infected roots and stem, and their similarity to the original isolates’ morphological characteristics was verified. The casual isolates that obeyed the criteria specified by Koch’s postulates were conducted for further study.

### 2.6. Effect of Ozone Treatment on the Development of C. pilosula Postharvest Disease

Healthy *C. pilosula* were treated and inoculated according to the above Section 2.5. Sterile water spraying inoculation was regarded as control. Gaseous ozone was generated by the OSAN ozone generator (Aoshan Huanbao Technology Industry Co., Ltd., Dalian, China). The concentration of ozone (2 mg L^−1^) was adjusted using an ozone detector [18]. Ozone fumigation was performed in a closed transparent bag (80 cm long × 60 cm wide) (25 °C, relative humidity 75%), and the inoculated samples were subjected to ozone treatment for 1 and 2 h each day, respectively. The treatment continued for 7 days. Subsequently, the treated tissue was stored for 56 days in plastic bags (22 ± 2 °C, 75–80%). The disease development was evaluated by statistically determining the disease index and natural incidence according to the report by Sha et al. [24] with minor modifications (Table 1). Each treatment contained three replicates, and one replicate included 50 samples.

Disease Index = [sum (class frequency × score of rating class)/[(Total number of plants) × (maximal disease index)] × 100;

Disease incidence = (Number of the infected plants/the number of plants sampled) × 100;

Class frequency: the number of diseased plants at each rate;

Score of rating class: the diseased value for each rate.

### 2.7. Effect of Ozone Treatment on the Mycotoxin Accumulation in the Rotten Tissue

For mycotoxins analysis, the samples treated by ozone fumigation were collected and the rotten tissue was excised from the diseased root and immediately kept in liquid nitrogen and stored at −80 °C until mycotoxin analysis. Each treatment contained three replicates, and one replicate included 50 samples.

A 5.0 g frozen sample was ground in liquid nitrogen, then was transferred to a 50 mL centrifuge tube with extraction solvent to extract mycotoxin. For different kinds of mycotoxin, various purification and detection methods were employed, patulin (PAT) purification and detection was carried out as described by the method [25]; trichothecene purification and detection was performed according to the method [26].

### 2.8. Statistical Analysis

The data of disease index, disease incidence, and mycotoxin accumulation were expressed as the means [standard error (±SE)] using Duncan’s multiple range tests. The experiment of the effect of ozone on postharvest disease and mycotoxin accumulation of *C. pilosula* was performed at least three times. Statistical analyses were performed using SPSS v.17.0 (SPSS, Inc., Chicago, IL, USA), and Duncan’s multiple range test (*p* < 0.05) was employed in this study.

## 3. Results

### 3.1. Disease Development of Freshly Harvested C. pilosula during Different Storage Stages

With the extension of storage time, the disease development of freshly harvested *C. pilosula* was more severe (Figure 1). When stored for 7 days, mild symptoms of disease were observed, and a small number of white hyphae and moldy spores were found on the main root and lateral root of *C. pilosula*. When stored for 14 days, hyphae were gradually diffused, and more white hyphae and moldy spores covered the root of *C. pilosula*. After 21 days of storage, colonies expanded and their color changed slightly from white to light green, with vigorous mycelium growth. When stored for 28 days, some yellow, red, and green hyphae appeared on the surface of the *C. pilosula*. When stored for 42 days, multiple colonies were distributed over the surface of *C. pilosula* and the tissue was damaged. After 56 days of storage, *C. pilosula* was seriously diseased, and the tissue was wrinkled, soft, and even rotten.

### 3.2. The Isolation of Pathogen from C. pilosula with Postharvest Disease during Different Storage Stages

During the whole storage period, a total of nine isolates of pathogenic fungi were isolated and purified. The nine different fungi were obtained using repeated plate streaking during different storage stages. For example, on the 7th day of storage, 2 isolates were obtained, which were named 7-1 and 7-2, respectively; on the 14th day of storage, 5 isolates were obtained, among them, 3 isolates were newly obtained, which were named 14-1, 14-2, and 14-3, respectively; on the 21st day of storage, 6 isolates were obtained, among them, a new isolate was named 21-1; on the 28th day of storage, 8 isolates were obtained, and two of them were newly isolated, named 28-1 and 28-2. There were no new isolates on the 42nd and 49th day. On the 56th day of storage, 9 isolates were obtained, and there was a new isolate named 56-1.

### 3.3. Morphological Identification of Pathogens at Different Storage Stages

After isolation and purification, the pathogens were cultivated on a PDA medium. Colony morphology, spore morphology, and sporangiophore morphology of the nine isolates of pathogenic fungi were observed.

For isolate 7-1, the mycelia grew fast at a rate of 24.57 mm/d on the PDA plate. The colony texture was flocculent with a white color, irregular edge, and no pigment (Figure 2A). Spores were spherical or nearly spherical in size at 4.8–5.6 μm × 5.0–6.2 μm (Figure 3A); sporangia were spherical with a diameter of 78–81 μm (Figure 4A). For isolate 7-2, the mycelia grew fast at a rate of 21.5 mm/d on the PDA plate. The colony texture was flocculent and white or gray-white with dense mycelia, a neat edge, and no pigment (Figure 2B); spores were spherical or subspherical and approximately 3.9–4.5 μm × 4.3–5.9 μm in size (Figure 3B); sporangia located at the end of hyphae were spherical, with a diameter of 59–63 μm (Figure 4B) (Table 2).

For isolate 14-1, the mycelia grew at a rate of 7.39 mm/d on the PDA plate. The colony was round or nearly round with a fluffy texture, neat or wavy edges, and a thick and dense mycelium; the secreted pigment was cinnamon colored, the surface and back of the edge were rose-red (Figure 2C). The number of conidia was small, the shape was oval or fusiform with a size of 2.4–3.0 μm × 3.0–3.6 μm (Figure 3C); 2 to 4 septa were observed, and the conidiophores had branches (Figure 4C). For isolate 14-2, the mycelia grew at a speed of 4.75 mm/d on the PDA plate. The colony texture was villous or cotton floc, with a light pink color, and the colony edge was light pink or white, with dense mycelia in the middle of the colony and sparse mycelia at the border of the colony (Figure 2D); conidia were oblong or fusiform and 2.1–3.2 μm × 2.8–3.5 μm in size (Figure 3D); conidiophores were erect and branched (Figure 4D). For isolate 14-3, the mycelia grew at a speed of 4.57 mm/d on the PDA plate. The colony was round with neat edges, pale purple or dark purple in color, and the edges were white or grayish-white with flocculent or fluffy hyphae (Figure 2E). The conidia with 3 to 4 septa were elongated oval, oval, or slightly curved, measuring 2.2–3.3 μm × 2.8–3.5 μm (Figure 3E); conidiophores had branches (Figure 4E).

For isolate 21-1, the mycelia grew at a speed of 4.29 mm/d on the PDA plate. The colony’s front color was white, while the back was pale yellow; mycelia were creeping and looser with a neat edge (Figure 2F); conidia were spherical or nearly spherical with a size of 2.3–3.2 μm × 2.8–3.5 μm (Figure 3F), and the conidiophores had branches (Figure 4F).

For isolate 28-1, the mycelia grew at a rate of 8.71 mm/d on the PDA plate. The front color of the colony was gray-green with dense granular texture, and the reverse was white (Figure 2G); conidia were striate, with colorless monospores, globose or oblate, with a size of 2.1–3.4 μm × 3.4–4.2 μm (Figure 3G); conidiophores were erect, septate, colorless, apex branched, and broom shaped (Figure 4G). For isolate 28-2, the mycelia grew with a speed of 6.50 mm/d on the PDA plate. A blue-green, and yellowish-brown with white areas were observed at the front and back of the colony, respectively. The center of the colony had protrusions and was slightly flocculent at the central surface and the texture was fluffy and powdery, forming several granular concentric rings (Figure 2H); Conidia were striate, spherical or ellipsoidal, with a size of 2.4–2.8 μm × 2.8–3.6 μm (Figure 3H); conidiophores were erect, septate, colorless, apex branched, and broom shaped (Figure 4H).

For isolate 56-1, the mycelia grew with a speed of 10.14 mm/d on the PDA plate. An orange-pink, and pink with annual rings was observed on the front and back of the colony (Figure 2I); the conidia were loosely gathered at the top of the mycelium, and were obovate or pear shaped, colorless, bicellular, with a size of 5.8–7.0 μm × 10–14 μm (Figure 3I); conidiophores were erect, colorless, with an ultimate swelling, and 2.0–3.5 μm × 100–160 μm in size (Figure 4I).

### 3.4. Molecular Identification of Pathogens at Different Storage Stages

Based on the above morphological observation during different storage stages, the nine isolates were further characterized by molecular biological technology based on the method of Gloria et al. [27]. The length of *ITS* primer amplified sequences were: 638 base pair (bp) of 7-1, 625 bp of 7-2, 538 bp of 14-1, 516 bp of 14-2, 533 bp of 14-3, 545 bp and 563 bp of 28-1, 761 bp of 28-2, and 589 bp of 56-1 (Figure 5A). The lengths of the sequences amplified by *TEF* primers were 695 bp for 14-1, 686 bp for 14-2, and 685 bp for 14-3 (Figure 5B). The lengths of the sequences amplified by *Bt* primers were 331 bp for 21-1, 448 bp for 28-1, 452 bp for 28-2, and 332 bp for 56-1 (Figure 5C). The sequences of the nine pathogens were searched using BLAST in NCBI, and the appropriate sequences were selected. The phylogenetic trees of *ITS*, *TUB*, and *TEF* were, respectively, constructed by MEGA7 (Figure 6).

The phylogenetic tree of *ITS* analysis (Figure 6A) revealed that isolate 7-1 was closely related to the strains of *JN943000.1*, *AM745431.1*, and *JN887460.1*, which were in the same branch, with a homology of 100%. The combination of the morphology and biological characters, isolate 7-1, was identified as *Actinomucor elegans*. Isolate 7-2 shared 100% homology with the strains of *MT626048.1*, *MT573485.1*, and *GU566234.1*, which were located in the same branch; therefore, based on morphology and biological characters, isolate 7-2 was identified as *Mucor hiemalis*. The phylogenetic tree of *ITS* analysis showed that 14-1 shared 100% homology with *MT525360.1*, *LC543657.1*, and *MK764994.1* in the same branch; thus, 14-1 was preliminarily identified as *Fusarium acuminatum*. Isolate 14-2 shared 100% homology with *KY365589.1*, *KY365574.1*, and *KY365564.1* in the same branch; therefore, 14-2 was preliminarily identified as *Fusarium equiseti*. Isolate 14-3 was closely related to the strains of *MT420651.1*, *MT420633.1*, and *MT420627.1*, which were located in the same branch, with a homology of 100%; thus, 14-3 was preliminarily identified as *Fusarium oxysporum*. Isolate 21-1 was identified as *Clonostachys rosea* on the same branch as *KX058045.1*, *KT921200.1*, and *KY365580.1*, with a homology of 100%. Isolate 28-1 and *KX243329.1*, *KT243328.1*, *MK578895.1* were in the same branch, with a homology of 98%; therefore, combination morphology and biological characters, 28-1, was identified as *Penicillium expansum*. Isolate 28-2 and *GU566234.1*, *FR670308.1*, *AY380455.1,* and MG228409.1 were in the same branch, with a homology of 98%; therefore, based on the morphology and biological characters, isolate 28-2 was identified as *Penicillium aurantiogriseum*. Isolate 56-1 and *MN372207.1*, *MN882763.1*, and *KY610499.1* were in the same branch, with a homology of 100%; therefore, based on the morphology and biological features, isolate 56-1 was identified as *Trichothecium roseum* (Figure 6A).

Based on *ITS* phylogenetic tree analysis, 14-1, 14-2, and 14-3 were the *Fusarium* species. In order to more correctly characterize the *Fusarium* species, the special primer of *TEF* for *Fusarium* was employed, and the *TEF* phylogenetic tree was constructed (Figure 6B). *TEF* phylogenetic tree analysis suggested that isolate 14-1 was closely related to *JX397842.1*, *KR108750.1*, and *JX397863.1*, which were in the same branch, with a homology of 100%. Isolate 14-2 was related to *KP639701.0*, *KT224199.1*, and *KT213298.1*, which were located in the same branch, with a homology of 100%. Isolate 14-3 was related with *MN417191.1*, *MN417185.1*, *MN417182.1*, and *MN417165.1*, which were in the same branch, with a homology of 100%. With the combined analyses of *ITS* and *TEF* phylogenetic trees and morphology feature, isolates 14-1, 14-2, and 14-3 were, respectively, identified as *Fusarium acuminatum*, *Fusarium equiseti*, and *Fusarium oxysporum* (Figure 6A,B). According to the results of the *ITS* and *TUB* phylogenetic tree, isolates 21-1, 28-1, 28-2, and 56-1 were identified as *Clonostachys rosea*, *Penicillium expansum*, *Penicillium aurantiogriseum*, and *Trichothecium roseum*, respectively (Figure 6A,C).

### 3.5. Pathogenicity Test

Pathogenicity tests were used to verify the pathogenicity of the isolates causing postharvest disease according to Koch’s postulates. During the whole incubation, for *C. pilosula* inoculated with 7-1 (*Actinomucor elegans*), dense white flocculent hyphae covered the main root on the 7th day, grew rapidly with time, and the expansion of colonies on the 28th day led to the decay of the epidermis of *C. pilosula* and leakage of sap (Figure 7A). For *C. pilosula* inoculated with 7-2 (*Mucor hiemalis*), gray flocculent hyphae appeared on the 7th day, grew rapidly on the 14th day, covering the whole tissue rotted on the 28th day (Figure 7B). For *C. pilosula* inoculated with 14-1 (*Fusarium acuminatum*): white dense mycelia were observed on the 7th day, the color of the dense mycelia gradually became rose-red on the 14th day; a rose-red glue was secreted and leaked from the *C. pilosula* tissue on 28th day (Figure 7C). For *C. pilosula* inoculated with 14-2 (*Fusarium equiseti*), there was less white velvet mycelial growth in the early stage, the colony became larger on the 14th day, and the color of *C. pilosula* epidermis turned pale pink or light yellow on the 28th day (Figure 7D). For *C. pilosula* inoculated with 14-3 (*Fusarium oxysporum*), there were white velvet colonies on the main root and lateral root on the 7th day, the color became light purple on the 14th day, and the color of *C. pilosula* epidermis turned dark purple on 28th day (Figure 7E). For *C. pilosula* inoculated with 21-1 (*Clonostachys rosea*), small white villous colonies were found on the surface of the tissue on the 7th day, the colony gradually expanded on the 14th day, and turned yellow and the tissue of *C. pilosula* produced yellowish glue on the 28th day (Figure 7F); For *C. pilosula* inoculated with 28-1 (*Penicillium expansum*), granular mold spots (2 to 3 mm in diameter) were observed on the surface of *C. pilosula* at the initial stage, a cluster of colonies gradually formed on the 14th day with a layer of blue powder on the surface; the plaque spread continuously, and the color of *C. pilosula* epidermis became gray and green, the tissue at the lesion developed soft rot and mildew on the 28th day (Figure 7G). For *C. pilosula* inoculated with 28-2 (*Penicillium aurantiogriseum*), small white granular mold spots appeared on the 7th day; the plaque expanded, and the number of colonies increased on the 14th day; the color of colonies turned blue and green, and *C. pilosula* developed serious mildew on the 28th day (Figure 7H).

For *C. pilosula* inoculated with 56-1 (*Trichothecium roseum*), white plaque with a diameter of 2 to 3 mm was initially observed on the 7th day; the orange-pink plaque expanded irregularly on the 14th day; the tissue was wrinkled with soft collapses and a dark color on the 28th day (Figure 7I). *C. pilosula* inoculated with the nine isolates had typical symptoms that were similar to the original natural symptoms. Based on the above typical and similar symptoms, the nine isolates were again identified through isolation, purification, and cultivation on PDA culture, and the same morphological and molecular biological characteristics were observed.

### 3.6. Ozone Fumigation Inhibited the Development of C. pilosula Postharvest Disease

The development of postharvest disease was effectively suppressed in *C. pilosula* inoculated with nine isolates after ozone fumigation treatment, and there was an ozone- exposure-time-dependent relationship with the inhibitory effect. For instance, the disease indexes in *C. pilosula* inoculated with *Actinomucor elegans* (7-1) after 1 and 2 h of ozone exposure were, respectively, 70% and 37% higher than those in control (Figure 8A). The disease incidences in *C. pilosula* inoculated with *F. acuminatum* (14-1) after 1 and 2 h ozone exposure were, respectively, 69% and 36% higher than those in control (Figure 8B). Similar results were obtained in the other isolates inoculated with *C. pilosula* after 1 and 2 h ozone exposure for 56 days of storage (Figure 8). It was obvious that with prolonged ozone treatment, the disease index and disease incidence dropped sharply.

### 3.7. Ozone Fumigation Inhibited the Mycotoxin Accumulation in the Rotten Tissue

More importantly, ozone treatment significantly inhibited mycotoxin accumulation in the rotten tissue of the inoculated *C. pilosula*. For instance, the content of patulin (PAT) of the rotten tissue in *C. pilosula* inoculated with *P. expansum* (28-1) was significantly (*p* < 0.05) inhibited by 38.9% and 53.0%, respectively, after 1 and 2 h of ozone exposure (Figure 9B). Similarly, the concentrations of 15-ADON and HT-2 of the rotten tissue in *C. pilosula* inoculated with *F. acuminatum* (14-1) were markedly decreased by 35.1% and 59.9% (15ADON), respectively, and 33.2% and 50.8 (HT-2), respectively, after 1 and 2 h ozone exposure (Figure 9A). Similar results were also found in *C. pilosula* inoculated with *T. roseum* (56-1) after ozone application (Figure 9C). For the control group and other pathogens infected with fresh *C. pilosula*, no mycotoxins were found (Figure 9).

## 4. Discussion

At present, numerous studies have addressed the pathogens causing preharvest disease in Chinese herbs, these reports indicated that *Fusarium* species are the dominant pathogens that cause preharvest disease in the various regions [28,29,30,31]. However, there are limited reports on postharvest diseases of freshly harvested Chinese herbal medicines during storage. Of relevance, Chen et al. [32] isolated and identified *P. crustosum*, *P. viridicatum*, *P. aurantiogriseum,* and *P. brevicompactum* from *Angelica sinensis* and *C. pilosula* decoction pieces during storage. Nevertheless, there are significant differences between Chinese medicinal decoction pieces and freshly harvested Chinese herbs as experimental materials. Chinese medicinal decoction pieces are mostly obtained from the Chinese herbal medicine market, and these products are available in dry or dehydrated forms. On the other hand, freshly harvested Chinese herbs come from the field, without any processing and treatments after harvest such as drying or dehydration. Therefore, fresh *C. pilosula* is more susceptible to molds and decay, owing to the higher water content and the abundance of nourishing substances that allow the growth of pathogens.

To our knowledge, this is the first report to isolate and characterize the pathogens causing postharvest disease of *C. pilosula* during different storage periods. The results indicated that, with the extension of storage time, the symptoms of the disease were more severe. Moreover, a total of 9 isolates were isolated and characterized based on the morphology and biological characters during storage for 56 days. All the nine isolates could cause different postharvest disease by Koch’s postulate. However, the results went beyond Koch’s postulate because Koch’s postulate is mainly applied to one pathogen. However, we initially observed in the postharvest disease that many pathogens were interacting for disease, and we isolated and identified the different pathogens, then inoculated the healthy *C. pilosula* with the obtained pathogen. Finally, we observed the different disease symptoms caused by different pathogens. Therefore, we thought Koch’s postulate had limitations for the results of this study.

*A. elegans* and *M. hiemalis* were the main pathogens causing mucor rot of freshly harvested *C. pilosula* on the 7th day of storage. After 14 days of storage, *Fusarium* species of *F. acuminatum*, *F. equiseti*, and *F. oxysporum* were the major pathogenic fungi causing root rot in *C. pilosula*. After 21 days of storage, besides the above-isolated pathogens, a new pathogen of *C. rosea* was isolated and characterized. On the 28th day of storage, blue mold caused by *Penicillum* spp. was the typical postharvest disease, and two new pathogens of *P. expansum* and *P. aurantiogriseum* were isolated and identified. On the 56th day of storage, a new pathogen *T. roseum* was isolated and characterized.

From the above results, the first postharvest disease was caused by *Mucor* infection on the 7th day of storage, and the main pathogens were *A. elegans* and *M. hiemalis*. *A. elegans* grew rapidly on PDA plates, and it took around 3 days to cover the whole petri dish. The morphology of colonies, spores and sporangiophores were similar to those isolated from necrotic skin lesions in humans, and the predominant fungal pathogen causing invasive *mucormycosis* for humans [33]. In addition, *A. elegans*, which can result in sufu’s white flake, was discovered in the sufu [34]. *Mucor hiemalis* had a fast growth speed on PDA medium, with a white colony initial color, which changed gradually to grayish brown (Figure 3B). The spore sacs were ellipsoid, and sporangiophores were erect and branched (Figure 4B), This observation was similar to the morphology of *M. hiemalis* causing mucor rot of mandarin fruit in California reported by Saito et al. [35]. The spores of the two Mucor species can survive in the air and infect hosts mainly through the air.

During storage of 14 days, *Fusarium* spp. were the main pathogens causing postharvest root rot of *C. pilosula*; among them, *F. acuminatum* had the strongest pathogenicity, and also produced mycotoxins, with nearly round colonies, neat edges, vigorous mycelia, flocculent, and dense (Figure 2C). Most conidia were fusiform and septate (Figure 3C), whose morphology was similar to *F. acuminatum* from garlic bulb rot in Serbia [36]. Moreover, *F. acuminatum* also metabolizes 15ADON and HT-2 toxins (Figure 9). The toxins 15-ADON and HT-2 are attributed to trichothecene with detrimental effects, including cytotoxicity, acute toxicity, immunotoxicity, and chronic toxicity, and pose a serious threat to human health. Tan et al. [37] suggested that DON and 3-ADON were detected in Fusarium head blight (FHB) of wheat infected by *F. acuminatum*. The colony of *F. equiseti* was white, and the mycelium had cotton-like flocculence (Figure 2D). Conidia were oval (Figure 3D), and the morphology was consistent with the observation by Afroz et al. [22], who mentioned that *F. equiseti* isolated from cabbage *Fusarium* wilt in Korea had a morphology of loosely floccose and whitish-brown aerial mycelia, and the pigmentation of pale orange on PDA medium; moreover, the size and morphology of macroconidia and chlamydospores were also in accordance with our observation. The *F. oxysporum* colony was initially white, then turned off-white later, to light purple-dark purple pigment. Hyphae were flocculent or villous and denser (Figure 2E), the conidia were elongated oval and slightly pointed at both ends (Figure 3E), which was consistent with the morphology of *F. oxysporum* isolated from sesame plants. Previous study revealed that *Fusarium* species is attributed to “latent infected pathogen” [38], the infection of *Fusarium* usually occur in the field (growth stage of plants) during the blooming or heading of plants. The spores adhere to the surface of the leaf, then transported into the calyx, finally colonize on the root of plants, and remain in a latent status until there are favorable environmental conditions (such as high temperature and humidity) [39]. Therefore, it is possible to manage plant disease more effectively during the pathogen infection time.

After 21 days of storage, *C. rosea* was isolated and identified, but its disease infection was not severe. The hyphae were white with neat edges, mycelium was creeping and loose (Figure 2F), and the morphology was similar to that isolated from root rot of *Astragalus membranaceus* in China [40]. *C. rosea* has been regarded as a typical biocontrol fungus to inhibit pathogenic fungal growth. The most serious postharvest disease was blue mold caused by *Penicillium* spp., including *P. expansum* and *P. aurantiogriseum*. *P. expansum* is a typical postharvest pathogen of pome fruits (such as apples and pears) causing blue mold and patulin contamination. The isolated *P. expansum* colony was grayish-green, and the edge was white, with a granular or velvet dense texture (Figure 2G). The microconidia were colorless with monospores that were globose or oblate. The number of conidia was high, and the conidiophores were erect, separated, and colorless (Figure 3G), whose morphology was consistent with the blue mold of apple fruits as reported by Bahri et al. [41]. In the present study, patulin (PAT) was detected in *C. pilosula* infected by *P. expansum*. Patulin is a secondary metabolite generated by *Aspergillus* and *Penicillium* species under favorable conditions, and usually found in blue mold disease of pome fruits and their corresponding products. The maximum limit of PAT in fruit juice was set as 50 μg/kg by the European Union [42]; however, no standard of PAT level limits is established in Chinese herbal medicines. *P. aurantiogriseum*, a kind of plant endophytic fungi, is also a casual pathogen causing fruit postharvest decay. Shim et al. [43] isolated the pathogen from pear fruit infected with blue mold in Korea, and Liu et al. [44] isolated *P. aurantiogriseum* from fresh-cut lettuce corruption. The cultivation characters and microscopic examination (Figure 2H, Figure 3H and Figure 4H) in our study were very similar to the above reports’ morphological characteristics. *Penicillium* species are attributed to “wound pathogens” [38] that mainly infect host plants through wounds sustained by fruit cracking or mechanical damage during harvesting, transportation, handling, and storage. Therefore, to avoid pathogen infection, delicate operations should be recommended during packing and handling. 

*T. roseum* was identified on the 56th day of storage, as the predominant fungus causing *C. pilosula* disease. *T. roseum* is a typical necuotrophic fungal pathogen that can infect various postharvest fruits and vegetables, leading to trichothecenes contamination. Sharma et al. [45] isolated *T. roseum* from postharvest pink rot of avocado in Israel. *T. roseum* is also known to cause pink rot in muskmelon and apple fruit. The cultivation characters and microscopic examination of the pathogen from the pink rot of fruit were consistent with our observation. Moreover, 15-ADON, DON, and HT-2 toxins were found in the rotten tissue of *C. pilosula* pink rot caused by *T. roseum*, which were trichothecenes. Tang et al. [46] also suggested T-2 toxin and neosolaniol (NEO) were detected in the core rot of apple fruit infected by *T. roseum*. *T. roseum* is attributed to “wound pathogen”, which mainly infects host plants through wounds; however, it can also carry out latent infection when the plant grows in the field, then end its colonization and development, without showing any symptoms until the plant is harvested. Therefore, it is a challenging job to manage the disease caused by *T. roseum*. The combination application of preharvest spray and postharvest treatment is proposed.

Ozone, a strong oxidant compound, is widely used to control postharvest decay. In the present study, ozone treatment significantly reduced the development of postharvest disease of freshly harvested *C. pilosula*, especially, when the herbs were exposed to 2 h treatment compared to 1 h. The result was in accordance with the report that ozone exposure for 10 min was more effective than 5 min in controlling pear disease [47]. The inhibitory effect of ozone was attributed to oxidation. Ozone can attack the cell plasma of plant pathogenic fungus, leading to membrane lipid peroxidation, and cell membrane integrity destruction, and loss of pathogenicity to the plant host [18]. In addition, we found that, as the ozone exposure time increased, the control effect became more visible. The reason is maybe that there is a cumulative effect for ozone exposure, the cumulative effect of 2 h ozone exposure was more than that of 1 h ozone exposure.

More importantly, we found that ozone treatment greatly suppressed the mycotoxin accumulation in the rotten tissue of the infected *C. pilosula* (Figure 9). A similar result was documented by Xue et al. [48], who indicated ozone application significantly reduced NEO accumulation. Considering the two reasons, ozone treatment controlled postharvest disease by inhibiting fungi growth; on the other hand, ozone can act directly with the chemical structure of mycotoxin, thus destroying the structure of mycotoxin [48].

## 5. Conclusions

A total of nine isolates were identified and characterized by morphology and molecular technology from postharvest diseases of *C. pilosula* during different storage stages. The nine isolates, with varying morphologies of colony, spore, and conidiophore, could cause several symptomatic postharvest diseases during different storage stages. Postharvest diseases of *C. pilosula* during storage can also metabolize and produce mycotoxins, which pose health threats to humans. Ozone application not only controlled postharvest diseases of *C. pilosula* during storage but also reduced mycotoxin accumulation. Therefore, in order to effectively control postharvest decay, with different pathogens having various infection modes and infection periods, the incorporation application of preharvest spray and postharvest treatment should be proposed; on the other hand, ozone application needs to be widely recommended and the mechanism of ozone action on disease and mycotoxins suppression should be further studied.

## Figures and Tables

**Figure 1 jof-09-00146-f001:**
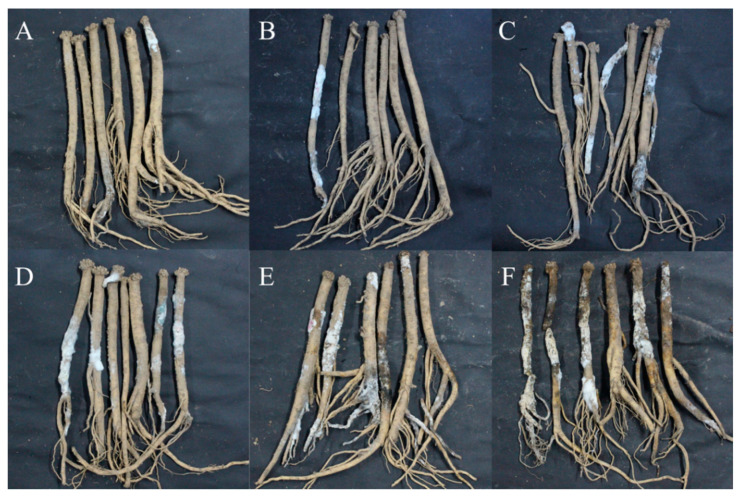
Disease development and naturally occurring symptoms of the fresh *C. pilosula* during different storage stages after harvest. (**A**) 7 d; (**B**) 14 d; (**C**) 21 d; (**D**) 28 d; (**E**) 42 d; (**F**) 56 d.

**Figure 2 jof-09-00146-f002:**
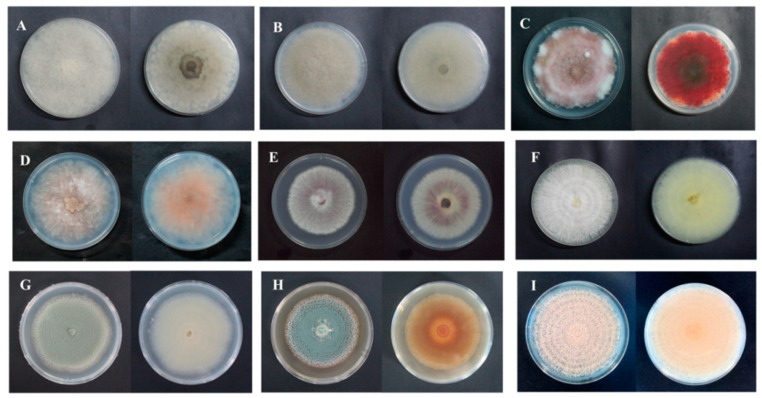
Colony morphology of isolates from fresh *C. pilosula* with postharvest disease during different storage stages. (**A**) *Actinomucor elegans*; (**B**) *Mucor hiemalis*; (**C**) *Fusarium acuminatum*; (**D**) *Fusarium equiseti*; (**E**) *Fusarium oxysporum*; (**F**) *Clonostachys rosea*; (**G**) *Penicillium expansum*; (**H**) *Penicillium aurantiogriseum*; (**I**) *Trichothecium roseum*.

**Figure 3 jof-09-00146-f003:**
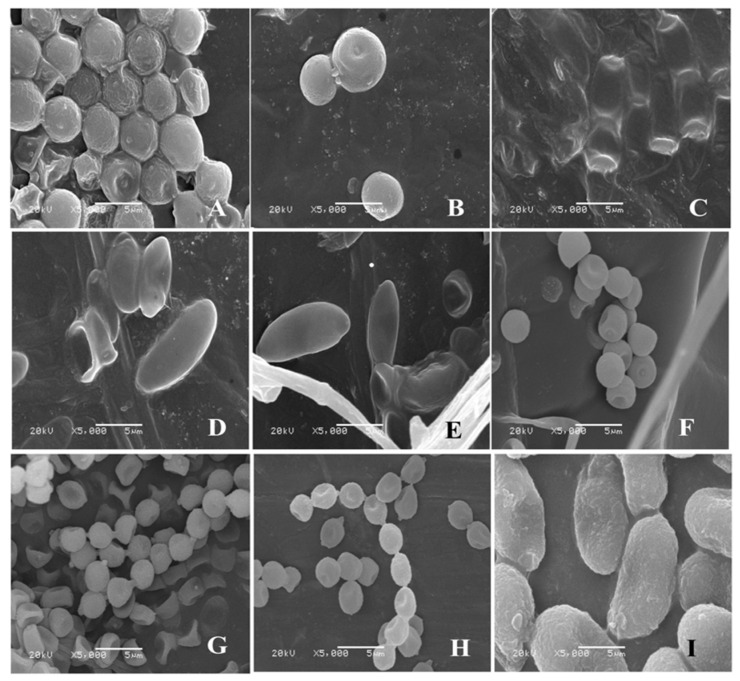
Morphology of conidium of isolates from fresh *C. pilosula* with postharvest disease during different storage stages. (**A**) *Actinomucor elegans*; (**B**) *Mucor hiemalis*; (**C**) *Fusarium acuminatum*; (**D**) *Fusarium equiseti*; (**E**) *Fusarium oxysporum*; (**F**) *Clonostachys rosea*; (**G**) *Penicillium expansum*; (**H**) *Penicillium aurantiogriseum*; (**I**) *Trichothecium roseum*.

**Figure 4 jof-09-00146-f004:**
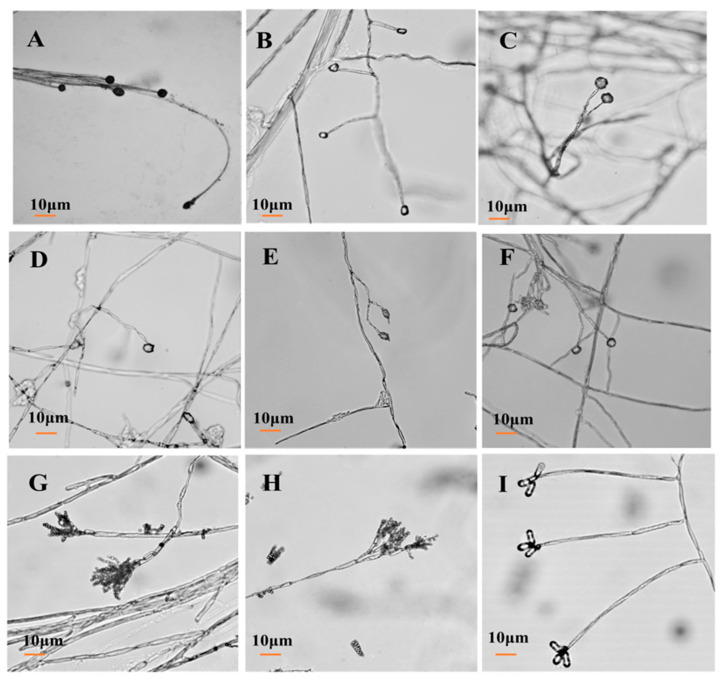
Morphology of conidiophore of isolates from *C. pilosula* with postharvest disease during different storage stages. (**A**) *Actinomucor elegans*; (**B**) *Mucor hiemalis*; (**C**) *Fusarium acuminatum*; (**D**) *Fusarium equiseti*; (**E**) *Fusarium oxysporum*; (**F**) *Clonostachys rosea*; (**G**) *Penicillium expansum*; (**H**) *Penicillium aurantiogriseum*; (**I**) *Trichothecium roseum*.

**Figure 5 jof-09-00146-f005:**
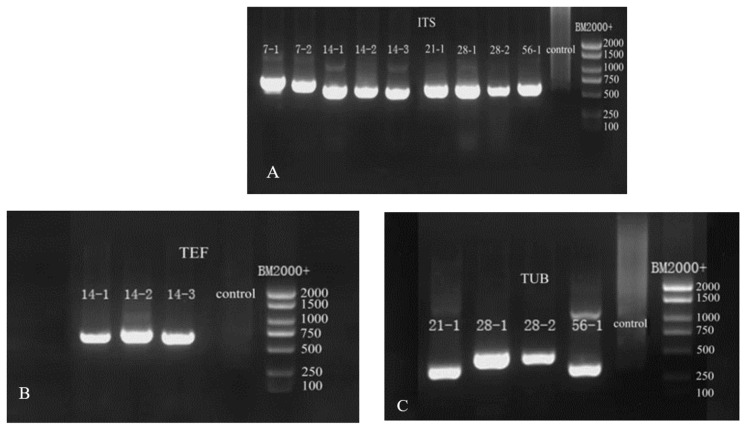
Gel electrophoresis images of PCR amplification products. (**A**) *ITS* gel electrophoresis images; (**B**) *TEF* gel electrophoresis images; (**C**) *TUB* gel electrophoresis images.

**Figure 6 jof-09-00146-f006:**
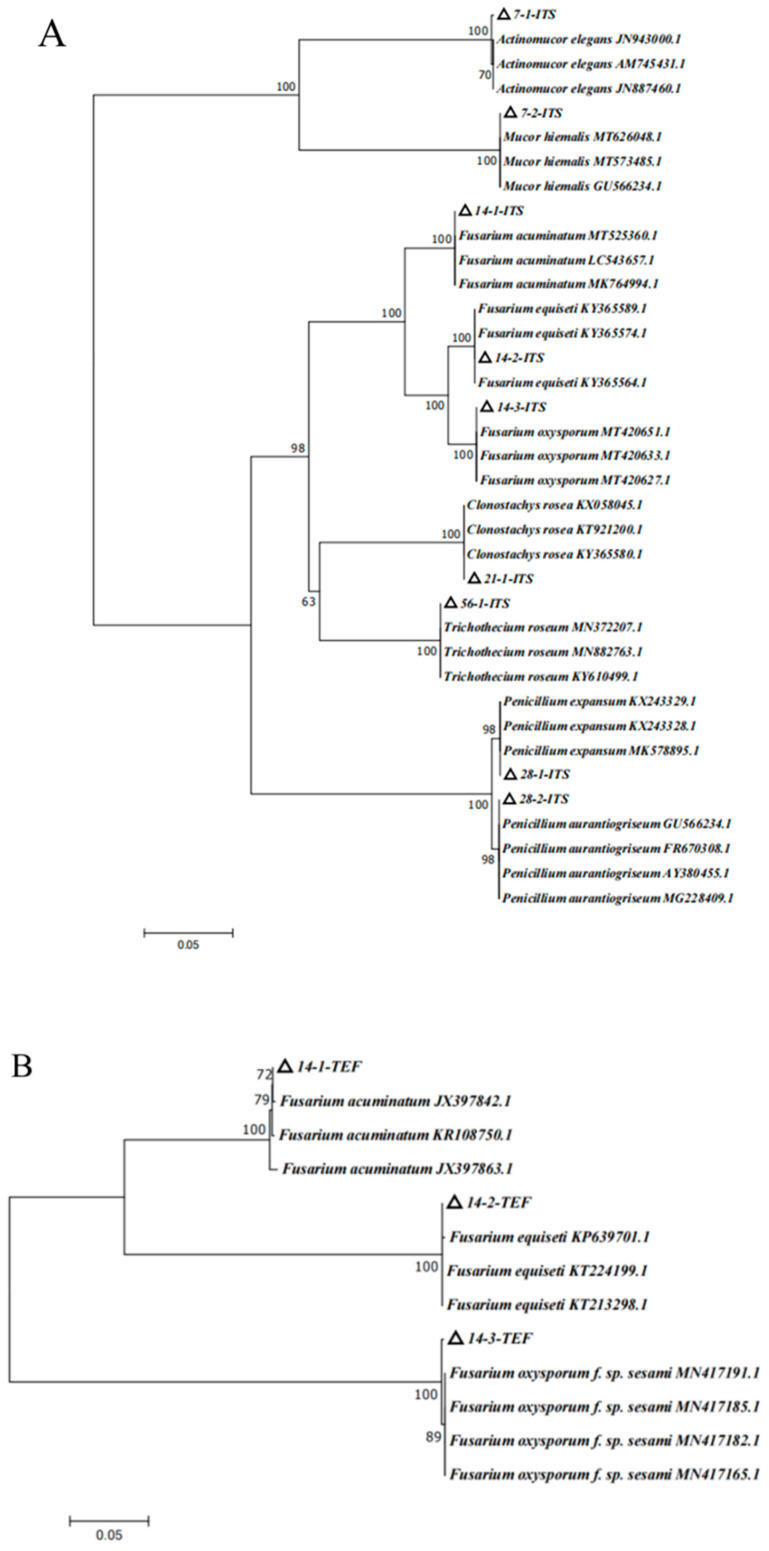
Phylogenetic tree of isolates based on analysis of different fungal genes. (**A**) *ITS* phylogenetic tree; (**B**) *TEF* phylogenetic tree; (**C**) *TUB* phylogenetic tree.

**Figure 7 jof-09-00146-f007:**
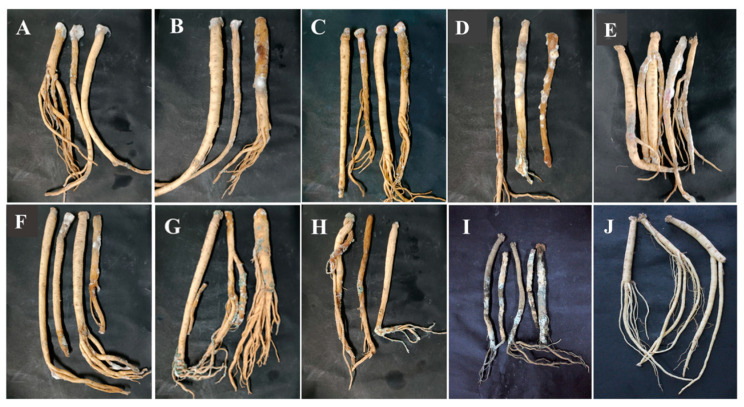
Pathogenicity testing of isolates from fresh *C. pilosula* with postharvest disease during different storage stages. (**A**) *Actinomucor elegans*; (**B**) *Mucor hiemalis*; (**C**) *Fusarium acuminatum*; (**D**) *Fusarium equiseti*; (**E**) *Fusarium oxysporum*; (**F**) *Clonostachys rosea*; (**G**) *Penicillium expansum*; (**H**) *Penicillium aurantiogriseum*; (**I**) *Trichothecium roseum*. (**J**) Control (no inoculation).

**Figure 8 jof-09-00146-f008:**
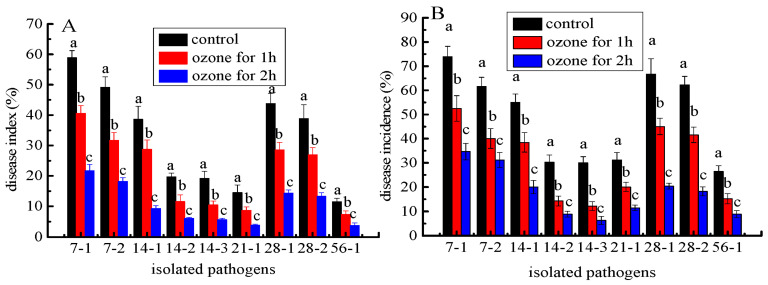
The effect of ozone treatment on disease index (**A**) and disease incidence (**B**) of fresh *C. pilosula* infected by the 9 isolates at 56 days of storage. (7-1: *Actinomucor elegans*; 7-2: *Mucor hiemalis*; 14-1: *Fusarium acuminatum*; 14-2: *Fusarium equiseti*; 14-3: *Fusarium oxysporum*; 21-1: *Clonostachys rosea*; 28-1: *Penicillium expansum*; 28-2: *Penicillium aurantiogriseum*; 56-1: *Trichothecium roseum*). The different letters indicate significant differences during the same storage period (*p* < 0.05).

**Figure 9 jof-09-00146-f009:**
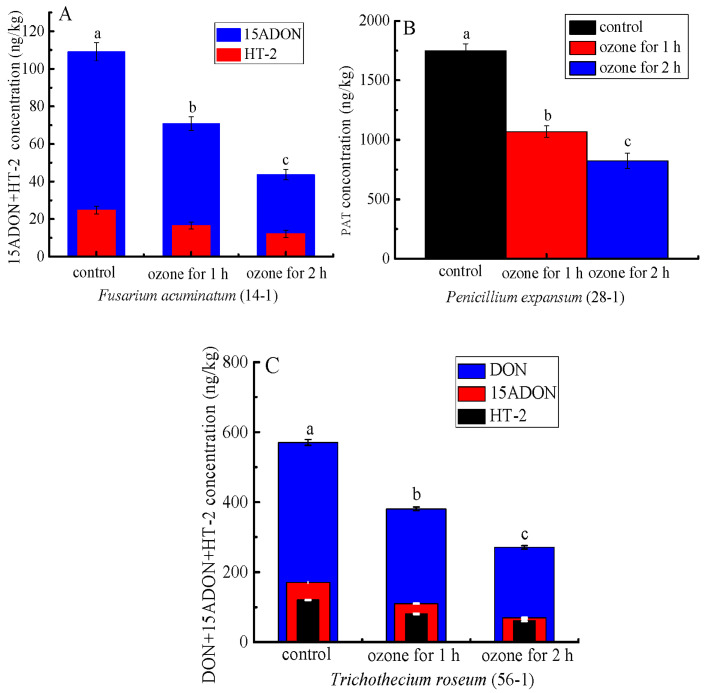
The effect of ozone treatment on mycotoxin accumulation of fresh *C. pilosula* infected by the isolates at 56 days of storage. (**A**) 15ADON and HT-2 toxin for *Fusarium acuminatum* (14-1); (**B**) Patulin (PAT) for *Penicillum expansum* (28-1); (**C**) DON, 15ADON, and HT-2 toxin for *Trichothecium roseum* (56-1); The different letters indicate significant differences during the same storage period (*p* < 0.05).

**Table 1 jof-09-00146-t001:** Disease classification standard.

Disease Rate	Symptom
0	No disease
1	The diseased area accounts for 1–5% of the total area of *C. pilosula*
2	The diseased area accounts for 6–25% of the total area of *C. pilosula*
3	The diseased area accounts for 25–50% of the total area of *C. pilosula*
4	The diseased area accounts for 51–75% of the total area of *C. pilosula*
5	The diseased area accounts for 76–100% of the total area of *C. pilosula*

**Table 2 jof-09-00146-t002:** Morphological characteristics of pathogens isolated at different storage periods.

Colony Morphology				Microscopic Morphology		
Strain Number	Front Color	Back Color	Texture	Growth Speed (mm/d)	Conidium	Conidiophore
7-1	White	White	Flocculent	24.57	Spherical or near spherical	Sporangium
7-2	Off-white	White	Flocculent	21.5	Spherical or near spherical	Sporangium
14-1	Rose-red	Rose-red	Fluffiness	7.39	Spindle shaped	Erect and branch
14-2	Light pink	Light pink	Fluffiness	4.75	Spindle shaped	Erect and branch
14-3	Dark purple	Dark purple	Fluffiness	4.57	Spindle shaped	Erect and branch
21-1	White	Light yellow	Fluffiness	4.29	Spherical or near spherical	Erect and branch
28-1	Grey-green	White	Powdery or grainy	8.71	Spherical or flat spherical	Erect, broom
28-2	Blue-green	Tan	Grainy	6.50	Spherical or flat spherical Pear shaped or obovate	Erect, broom
56-1	Orange	Orange	Grainy	10.14		Erect

## Data Availability

The data presented in this study are included in the article; further inquiries can be directed to the corresponding author.

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
