# Peer review of "Isolation of Main Pathogens Causing Postharvest Disease in Fresh Codonopsis pilosula during Different Storage Stages and Ozone Control against Disease and Mycotoxin Accumulation"

_jof, 2023, doi:10.3390/jof9020146_

Round 1

Reviewer 1 Report

The paper with the title “Isolation of Main Pathogens Causing Postharvest Disease in Fresh Codonopsis pilosula during Different Storage Stages and Ozone Control against Disease and Mycotoxin Accumulation”, the authors study the causal agents for the postharvest disease of fresh C. pilosula. In addition, the control of ozone was examined against the isolates and mycotoxin accumulation. The results are acceptable and meet the journal scope. Below are found some suggestions and comments for authors for further improvement of the manuscript.

1- The manuscript needs English editing.

2- The scientific names of fungi should be revised and written in italic.

3- Line 46-47, “Z.Y. Zhao, Sch., Ziling” …. Not written in italics

4- Although the authors identified fungal isolates using molecular techniques, …. They didn’t deposit the obtained sequence in Genebank and didn’t provide the accession number.

5- Fig. 4, authors should provide the morphology of fungal isolates with higher magnification power.

6- Fig. 6, unclear, please magnifies the image.

7- Line 415-417, please revise, isn’t clear, and there is a miswriting in the provided value (070). Also, please provide the obtained values for the effect of ozone on the disease development as percent to be clear for the readers. 

Author Response

Dear Reviewer,

On behalf of our co-authors, we thank you very much for giving us this opportunity to revise our manuscript, we appreciate editor and reviewers very much for your positive and constructive comments and suggestions on our manuscript entitled “Isolation of Main Pathogens Causing Postharvest Disease in Fresh Codonopsis pilosula during Different Storage Stages and Ozone Control against Disease and Mycotoxin Accumulation”. (Manuscript ID: jof-2153731).

We have studied carefully reviewer’s comments and have made revision which marked up using the “Track Change” in the revised manuscript. We asked Dr. William Oyom, whose mother language is English, to edit the English grammar and sentence, and improve the English writing level. We also tried our best to revise our manuscript according to the comments. The revised manuscript was attached, and responses to the specific comments are detailed below.

We would like to express our great appreciation to you and reviewers for comments on our manuscript. Looking forward to hearing from you.

Thank you and best regards.

Sincerely yours,

Prof. Dr. Huali Xue

Gansu Agricultural University

Reviewer comments:

Reviewer: 1

The paper with the title “Isolation of Main Pathogens Causing Postharvest Disease in Fresh Codonopsis pilosula during Different Storage Stages and Ozone Control against Disease and Mycotoxin Accumulation”, the authors study the causal agents for the postharvest disease of fresh C. pilosula. In addition, the control of ozone was examined against the isolates and mycotoxin accumulation. The results are acceptable and meet the journal scope. Below are found some suggestions and comments for authors for further improvement of the manuscript.

  1. The manuscript needs English editing.

Response: Thank you very much your good suggestion. We asked Dr. William Oyom whose mother language is English to edit the English grammar and sentence, and improve the English writing level, we hope the English writing can meet the requirement of “Journal of Fungi”. 

  1. The scientific names of fungi should be revised and written in italic.

Response: Thanks for your good suggestion. We have already checked and modified them.

Line 2, “Codonopsis pilosula” was replaced with “Codonopsis pilosula

Line 1123-1125: “The morphology of colony, spores and sporangiophores were similar with the one isolated from necrotic skin lesion in human, it is the predominant fungal pathogen causing invasive mucormycosis for humans [33].” has been replaced with “The morphology of colony, spores and sporangiophores were similar to those isolated from necrotic skin lesions in humans, and the predominant fungal pathogen causing invasive mucormycosis for humans [33].”

Line 1430-1431: “Moreover, 15-ADON, DON, HT-2 toxin were found in the rotten tissue of C. pilosula pink rot caused by T. roseum, which are trichothecenes.” has been replaced with “Moreover, 15-ADON, DON, and HT-2 toxin were found in the rotten tissue of C. pilosula pink rot caused by T. roseum, which were trichothecenes.”

Line 1773-1774, reference 16: “Identification of Acremonium acutatum and Trichothecium roseum isolated from grape with white stain symptom in Korea.” has been replaced with “Identification of Acremonium acutatum and Trichothecium roseum isolated from grape with white stain symptom in Korea.”

Line 1781, reference 19: “Survey for toxigenic Fusarium species on maize kernels in China.” has been replaced with “Survey for toxigenic Fusarium species on maize kernels in China.”

Line-1778-1789, reference 22: “First report of Fusarium wilt caused by Fusarium equiseti on cabbage (Brassica oleracea var. capitate) in Korea.” has been replaced with “First report of Fusarium wilt caused by Fusarium equiseti on cabbage (Brassica oleracea var. capitate) in Korea.”

Line-1791-1792, reference 23: “First report of Colletotrichum black leaf spot on strawberry caused by Colletotrichum Siamense in China.” has been replaced with “First report of Colletotrichum black leaf spot on strawberry caused by Colletotrichum Siamense in China.”

Line 1827-1828, reference 39: “Assessment of Fusarium infection and mycotoxin contamination of wheat kernels and flour using hyperspectral imaging.” has been replaced with “Assessment of Fusarium infection and mycotoxin contamination of wheat kernels and flour using hyperspectral imaging.”

Line-1832-1833, reference 41: “First report disease of Clonostachys rosea causing root rot on Astragalus membranaceus in China.” has been replaced with “First report disease of Clonostachys rosea causing root rot on Astragalus membranaceus in China.”

  1. Line 46-47, “Z.Y. Zhao, Sch., Ziling” …. Not written in italics

Response: Thanks for your good suggestion. We have already checked and modified, the sentence of “Using molecular technology, Yu et al. [9] isolated and identified five pathogens in China: Puccinia Campanumoeae Pat causing rust disease, Helicobasidium mompa Tanaka causing violet root rot, Sphaerotheca Codonopisi (Golov.) Z.Y. Zhao causing powdery mildew, Fusarium oxysporum Schl. causing root rot, and Septoria codonopsidis Ziling causing blight of Codonopsis tangshen in Chongqing.” has been replaced with “Using molecular technology, Yu et al. [9] isolated and identified five pathogens in China: Puccinia Campanumoeae Pat causing rust disease, Helicobasidium mompa Tanaka causing violet root rot, Sphaerotheca Codonopisi (Golov.) Z.Y. Zhao causing powdery mildew, Fusarium oxysporum Schl. causing root rot, and Septoria codonopsidis Ziling causing blight of Codonopsis tangshen in Chongqing.” in the revised manuscript.

  1. Although the authors identified fungal isolates using molecular techniques, …. They didn’t deposit the obtained sequence in Genebank and didn’t provide the accession number.

Response: Thanks for your good suggestion. Although we identified fungal isolates, we are so sorry for our not deposit the obtained sequence in Genebank. If you think it is very necessary, then we can supply the sequences of ITS, TEF and TUB in supporting information in the manuscript. The sequences of ITS, TEF and TUB are following that:

ITS

>7-1-ITS

TGGGCCTACGGGTTTGGTTTTTCTCTTATTTTTTACCGTGAACTGTCTTATAGCATGGCGCTAGTAGAGATGCCTGAGCCACCATACGGGGTAGGCGGCACAGGATGATTTTAATCGAAGCCATGGTCAAGCCGACTTTTTTTCAGCTTGGTACCCCAAAAATTAATTATTCTACCAAATGAATTCAGTATTAATATTGTAACATGGGCTCGCTGAAAGGTGGCCTATAAAACAACTTTTAACAACGGATCTCTTGGTTCTCGCATCGATGAAGAACGTAGCAAAGTGCGATAACTAGTGTGAATTGCATATTCAGTGAATCATCGAGTCTTTGAACGCATCTTGCACCTGCTGGTATTCCAGCAGGTACGCCTGTTTCAGTATCAGAAACAACTCTTCCCCTAAGATTTTTTTGAATCATAGGGGGACATTGAGGGTATCTGGCTTAGAAGTAAAATCTCTAGCCCGGAGACGCTTTAAATGACTTAAGGCCTGCAAGCCAAAGTTTGATTGCGCCTGAACTTTTTCTTAATTTCAAGCGAAAGCTCTTGCGAACTAGAACTTTATTATTGCCTTGGGGGCCTCCCAAAGAAAACTTTCAACAACTTGATCTGAAATCAGGTGGGATTACCCGCTGA

>7-2-ITS

AACCTGCGGAAGGATCATTAAATAATTTAGATGGCCTTTGCTAGTTTTCTAGCGAATGGTTCATTCTTTTTTACTGTGAACTGTTTTAATTTTTCAGCGTCTGAGGAATGTCTTTTAGCCATAGGGATAGGCTACTAGAATGTTAACCGAGCTGAAAGTCAGGCTTAGGCCTGGTATCCTATTAATTATTTACCAAAAGAATTCAGTATTATAATTGTAACATAAGCGTAAAAAACTTATAAAACAACTTTTAACAACGGATCTCTTGGTTCTCGCATCGATGAAGAACGTAGCAAAGTGCGATAACTAGTGTGAATTGCATATTCAGTGAATCATCGAGTCTTTGAACGCAACTTGCGCTCAATGGTATTCCATTGAGCACGCCTGTTTCAGTATCAAAAACACCCCACATTCATAATTTTGTTGTGAATGGAAATGAGAGTTTCGGCTTTATTGCTGAATTCTTTAAAATTATTAGGCCTGAACTATTGTTCTTTCTGCCTGAACATTTTTTTAATATAAAGGAATGCTCTAGTAAAAAGACTATCTCTGGGGCCTCCCAAATAAATCATTCTTAAATTTGATCTGAAATCAGGCGGGATTACCCGCTGAACTTAAGCATATC

>14-1-ITS

CTGCGGAGGGATCATTACCGAGTTTACAACTCCCAAACCCCTGTGAACATACCTTAATGTTGCCTCGGCGGATCAGCCCGCGCCCCGTAAAACGGGACGGCCCGCCAGAGGACCCAAACTCTAATGTTTCTTATTGTAACTTCTGAGTAAAACAAACAAATAAATCAAAACTTTCAACAACGGATCTCTTGGTTCTGGCATCGATGAAGAACGCAGCAAAATGCGATAAGTAATGTGAATTGCAGAATTCAGTGAATCATCGAATCTTTGAACGCACATTGCGCCCGCTGGTATTCCGGCGGGCATGCCTGTTCGAGCGTCATTTCAACCCTCAAGCCCCCGGGTTTGGTGTTGGGGATCGGCTCTGCCCTTCTGGGCGGTGCCGCCCCCGAAATACATTGGCGGTCTCGCTGCAGCCTCCATTGCGTAGTAGCTAACACCTCGCAACTGGAACGCGGCGCGGCCATGCCGTAAAACCCCAACTTCTGAATGTTGACCTCGGATCAGGTAGGAATACCCGCTGAACTTAAGCATATCA

>14-2-ITS

CTGCGGAGGGATCATTACCGAGTTTACAACTCCCAAACCCCTGTGAACATACCTATACGTTGCCTCGGCGGATCAGCCCGCGCCCCGTAAAACGGGACGGCCCGCCCGAGGACCCTAAACTCTGTTTTTAGTGGAACTTCTGAGTAAAACAAACAAATAAATCAAAACTTTCAACAACGGATCTCTTGGTTCTGGCATCGATGAAGAACGCAGCAAAATGCGATAAGTAATGTGAATTGCAGAATTCAGTGAATCATCGAATCTTTGAACGCACATTGCGCCCGCCAGTATTCTGGCGGGCATGCCTGTTCGAGCGTCATTTCAACCCTCAAGCTCAGCTTGGTGTTGGGACTCGCGGTAACCCGCGTTCCCCAAATCGATTGGCGGTCACGTCGAGCTTCCATAGCGTAGTAATAATACACCTCGTTACTGGTAATCGTCGCGGCCACGCCGTAAAACCCCAACTTCTGAATGTTGACCTCGGATCAGGTAGGAATACCCGCTGAACTTAAGCAT

>14-3-ITS

ACCTGCGGAGGGATCATTACCGAGTTTACAACTCCCAAACCCCTGTGAACATACCACTTGTTGCCTCGGCGGATCAGCCCGCTCCCGGTAAAACGGGACGGCCCGCCAGAGGACCCCTAAACTCTGTTTCTATATGTAACTTCTGAGTAAAACCATAAATAAATCAAAACTTTCAACAACGGATCTCTTGGTTCTGGCATCGATGAAGAACGCAGCAAAATGCGATAAGTAATGTGAATTGCAGAATTCAGTGAATCATCGAATCTTTGAACGCACATTGCGCCCGCCAGTATTCTGGCGGGCATGCCTGTTCGAGCGTCATTTCAACCCTCAAGCACAGCTTGGTGTTGGGACTCGCGTTAATTCGCGTTCCTCAAATTGATTGGCGGTCACGTCGAGCTTCCATAGCGTAGTAGTAAAACCCTCGTTACTGGTAATCGTCGCGGCCACGCCGTTAAACCCCAACTTCTGAATGTTGACCTCGGATCAGGTAGGAATACCCGCTGAACTTAAGCATATCAATAAGCGGAGGA

>21-1-ITS

CCTGCGGAGGGATCATTACCGAGTTTACAACTCCCAAACCCATGTGAACATACCTACTGTTGCTTCGGCGGGATTGCCCCGGGCGCCTCGTGTGCCCCGGATCAGGCGCCCGCCTAGGAAACTTAATTCTTGTTTTATTTTGGAATCTTCTGAGTAGTTTTTACAAATAAATAAAAACTTTCAACAACGGATCTCTTGGTTCTGGCATCGATGAAGAACGCAGCGAAATGCGATAAGTAATGTGAATTGCAGAATTCAGTGAATCATCGAATCTTTGAACGCACATTGCGCCCGCCAGTATTCTGGCGGGCATGCCTGTCTGAGCGTCATTTCAACCCTCATGCCCCTAGGGCGTGGTGTTGGGGATCGGCCAAAGCCCGCGAGGGACGGCCGGCCCCTAAATCTAGTGGCGGACCCGTCGTGGCCTCCTCTGCGAAGTAGTGATATTCCGCATCGGAGAGCGACGAGCCCCTGCCGTTAAACCCCCAACTTTCCAAGGTTGACCTCAGATCAGGTAGGAATACCCGCTGAACTTAAGCATATCA

>28-1-ITS

TGCGGAAGGATCATTACCGAGTGAGGGCCCTTTGGGTCCAACCTCCCACCCGTGTTTATTTACCTCGTTGCTTCGGCGGGCCCGCCTTAACTGGCCGCCGGGGGGCTCACGCCCCCGGGCCCGCGCCCGCCGAAGACACCCCCGAACTCTGCCTGAAGATTGTCGTCTGAGTGAAAATATAAATTATTTAAAACTTTCAACAACGGATCTCTTGGTTCCGGCATCGATGAAGAACGCAGCGAAATGCGATACGTAATGTGAATTGCAAATTCAGTGAATCATCGAGTCTTTGAACGCACATTGCGCCCCCTGGTATTCCGGGGGGCATGCCTGTCCGAGCGTCATTGCTGCCCTCAAGCCCGGCTTGTGTGTTGGGCCCCGTCCTCCGATTCCGGGGGACGGGCCCGAAAGGCAGCGGCGGCACCGCGTCCGGTCCTCGAGCGTATGGGGCTTTGTCACCCGCTCTGTAGGCCCGGCCGGCGCTTGCCGATCAACCCAAATTTTTATCCAGGTTGACCTCGGATCAGGTAGGGATACCCGCTGAACTTAAGCATATCAATAGG

>28-2-ITS

TTGTGAAAGTTTTAAATAATTTATATTTTCACTCAGACTTCAATCTTCAGACAGAGTTCGAGGTGTCTTCGGCGGCCGCGGGGGCTGAAGCCCCCCGCGCCAGTAAAGCGGCCCGCCGAAGCAACAAGGTAAAATAAACACGGGGAGGTTGGACCCAAAGGGCCCTCACTCGGTAATGACTTCCGTAGGGTTACCTGCGGAAGGATCATTACCGAGTGAGGGCCCTTTGGGTCCAACCTCCCACCCGTGTTTATTTTACCTTGTTGCTTCGGCGGGCCCGCCTTTACTGGCCGCCGGGGGGCTTCACGCCCCCGGGCCCGCGCCCGCCGAAGACACCCTCGAACTCTGTCTGAAGATTGAAGTCTGAGTGAAAATATAAATTATTTAAAACTTTCAACAACGGATCTCTTGGTTCCGGCATCGATGAAGAACGCAGCGAAATGCGATACGTAATGTGAATTGCAAATTCAGTGAATCATCGAGTCTTTGAACGCACATTGCGCCCCCTGGTATTCCGGGGGGCATGCCTGTCCGAGCGTCATTGCTGCCCTCAAGCCCGGCTTGTGTGTTGGGCCCCGTCCTCCGATTCCGGGGGACGGGCCCGAAAGGCAGCGGCGGCACCGCGTCCGGTCCTCGAGCGTATGGGGCTTTGTCACCCGCTCCGTAGGCCCGGCCGGCGCTTGCCGATCAACCCAAATTTTTATCCAGGTTGACCTCGGATCAGGTAGGGATACCCGCTGAACTTAAGCATATCAATAAGG

>56-1-ITS

TTTTGATATGCTTAAGTTCAGCGGGTATTCCTACCTGATCCGAGGTCAACCTTGGTAAAAGTTTGGGGTTTTACGGCGTGGCCGCGCCGCCTTCCGCCCGCGAGGTTTGTGCTACTACGCGGAGGAGGCAGCGGCGAGGCCGCCACTGGGTTTCGGGGACAGGCGCCCTTGGAGGACGCCTGGGCCCCAACGCCGGGCCCGCCCCCTCCCCCGCGAGGGGAAAAGGGGGGGGCCCGAGGGTTGAAATGACGCTCGGACAGGCATGCCCGCCAGAATACTGGCGGGCGCAATGTGCGTTCAAAGATTCGATGATTCACTGAATTCTGCAATTCACATTACTTATCGCATTTCGCTGCGTTCTTCATCGATGCCAGAACCAAGAGATCCGTTGTTGAAAGTTTTGATTTGTTTTGTTGCCTTTCGGCTCGCTCAGAGGATACACATGTTTGTTAAACAGGGTTCGTGTGGTCCCCGGCGGGCGCCTTGGGTCCGGGGCACGCAGCGCCCGGGGCGGTCCGCCGAAGCAACGGTAGGTAAGGTTCACAAAGGGTTGGGAGTTGTTTTGTTAACTCTATAATGATCCCTCCGCTGGTTCACCAACGGAGACCTTGTTACGCT

TEF

>14-1-TEF

GACAAGACTCACCTTAACGTCGTCGTCATCGGCCACGTCGACTCTGGCAAGTCGACCACTGTAAGTACAACCAACAACGGGTTGCTTATCTGCACTCGGAATCCGCCAAACCTGGCGGGGTATCACCAAAACATTTTGCTAACTTTTGACAGACCGGTCACTTGATCTACCAGTGCGGTGGTATCGACAAGCGAACCATCGAGAAGTTCGAGAAGGTTAGTCAATATCCCTTCGATTACGCGCGCTCCCGTCGATTCCCACGATTCGCTCCCTCACTCGAAACACATCCATTACCCCGCTCGAGTCCGAAAATTTTGCGGTGCGACCGTGATTTTTTCTGGTGGGGTATCTTACCCCGCCACTCGAGTCACGGATGCGCTTGCCCTGTTCCCACAAAACCTTACCACCCTGTCGCGCACTACATGTCTTGCAGTCACTAACCACTGGACAATAGGAAGCCGCCGAGCTCGGAAAGGGTTCCTTCAAGTACGCCTGGGTTCTTGACAAGCTCAAAGCCGAGCGTGAGCGTGGTATCACCATTGATATCGCTCTCTGGAAGTTCGAGACTCCTCGCTACTATGTCACCGTCATTGGTATGTTGTCACTGTCTCACACTATCATGTATTCATCATGCTAACATCTCTCTCAGATGCCCCCGGTCATCGTGATTTCATCAAGAACATGATCACTGGTAC

>14-2-TEF

CAAGACTCACCTTAACGTCGTCGTCATCGGCCACGTCGACTCTGGCAAGTCGACCACTGTGAGTACTACCCTCAATGACCTGCTTATCAGCAGTCATCAACCCCGCCATACGTGGTGGGGTAATATCAACTTACACATTTGCTGACAAAATTGCATAGACCGGTCACTTGATCTACCAGTGCGGTGGTATCGACAAGCGAACCATCGAGAAGTTCGAGAAGGTTGGTTTACATTTTCCTCGATCGCACGCCCTCTGCACACCGATCCATCACTCGAATCAGCCTCGATGACTGAATATGCGCCTGTCACCCCGCTCGAATACAAAATTTTGCGGTTCAACTGCAATTTTTTGGTGGGGCTCATACCCCGCTGCTCGAGTGACAGGCGCTTGCCCTCTTCCCACAAAATCACCTCTTGCGCGTCACGTGTCAATCAGTCACTAACCACCCGACAATAGGAAGCCGCCGAGCTCGGTAAGGGTTCCTTCAAGTACGCTTGGGTTCTTGACAAGCTCAAGGCCGAGCGTGAGCGTGGTATCACCATCGATATCGCTCTCTGGAAGTTCGAGACTCCTCGCTACTATGTCACCGTCATTGGTATGTTGTCATCACTTACACTCATGACCTTCTCATGCTAACATGTTTTCCAGACGCTCCCGGTCACCGTGATTTCATCAAGAACATGAT

>14-3-TEF

AAGACTCACCTTAACGTCGTCGTCATCGGCCACGTCGACTCTGGCAAGTCGACCACTGTGAGTACTCTCCTCGACAATGAGCATATCTGCCATCGTCAATCCCGACCAAGACCTGGCGGGGTATTTCTCAAAGTCAACATACTGACATCGTTTCACAGACCGGTCACTTGATCTACCAGTGCGGTGGTATCGACAAGCGAACCATCGAGAAGTTCGAGAAGGTTAGTCACTTTCCCTTCAATCGCGCGTCCTTTGCCCATCGATTTCCCCTACGACTCGAAACGTGCCCGCTACCCCGCTCGAGACCAAAAGTTTTGCAATATGACCGTAATTTTTTTGGTGGGGCACTTACCCCGCCACTTGAGCGACGGGAGCGTTTGCCCTCTTACCATTCTCACAACCTCAATGAGTGCGTCGTCACGTGTCAAGCAGTCACTAACCATTCAACAATAGGAAGCCGCTGAGCTCGGTAAGGGTTCCTTCAAGTACGCCTGGGTTCTTGACAAGCTCAAGGCCGAGCGTGAGCGTGGTATCACCATCGATATTGCTCTCTGGAAGTTCGAGACTCCTCGCTACTATGTCACCGTCATTGGTATGTTGTCGCTCATGCTTCATTCTACTTCTCTTCGTACTAACATATCACTCAGACGCTCCCGGTCACCGTGATTTCATCAAGAACATGATC

TUB

>21-1-Bt

GGTGCTGCTTTCTGGCAGACCATCTCTGGCGAGCACGGCCTCGACAGCAATGGTGTCTACAGCGGCACCTCCGAGCTCCAGCTCGAGCGCATGAACGTCTACTTCAACGAGGTATGTCCAGACCGAGCTTGACATATTCTGGTGATTTTCATCCTCTGACCGAGTTTTGGGTATAGGCCTCTGGCAACAAGTATGTTCCTCGCGCTGTCCTCGTCGATCTTGAGCCCGGTACCATGGATGCCGTCCGTGCTGGACCTTTTGGCCAGCTCTTCCGCCCCGACAACTTCGTTTTCGGCCAGTCTGGTGCTGGCAACAACTGGGCCAAGGGTCA

>28-1-Bt

GCTGCTTTCTGGTAAGTGCCGAGCTTTTTTTTCGCGTTGGGTATCAATTGACAATTTACTAACTGGATTGCAGGCAAACCATCTCTGGCGAGCACGGTCTCGATGGTGATGGACAGTAAGTTCAACGGTGATGGGTTTCTAGTAGATCACACGTCTGATATCTTGCTAGGTACAATGGTACCTCCGACCTCCAGCTCGAGCGTATGAACGTCTACTTCAACCATGTGAGTACACCGACTGTTTACCGAATAATCGTGCATCATCTGATCGGATCTTTTTCTTTGATAATCTAGGCCAGCGGTGACAAGTACGTTCCCCGTGCCGTTCTCGTCGATTTGGAGCCCGGTACCATGGACGCTGTCCGCTCCGGTCCCTTCGGCAAGCTTTTCCGCCCCGACAACTTCGTCTTCGGTCAGTCCGGTGCTGGTAACAACTGGGCCAAGGGTCA

>28-2-Bt

GCTGCTTTCTGGTAAGTACCGAGCTTTTTTTCCCCTTCGCGTTGGGTACAATTGACAGGTTACTAACTCGATTACAGGCAAACCATCTCTGGCGAGCACGGTCTCGATGGCGATGGACAGTAAGTTCTAATGGTGATGGGGGTTTCCGGTAGATCACACCTCTGATATCTTGCTAGGTACAATGGTACCTCCGACCTCCAGCTCGAGCGTATGAACGTCTACTTCAACCATGTGAGTGCAACGACTGGAAACCGAATAATCGTGCATCATCTGATCAGATGTTTTTCTTTGATAATCTAGGCCAGCGGTGACAAGTACGTTCCCCGTGCCGTTCTCGTCGATTTGGAGCCCGGTACCATGGACGCTGTCCGCTCCGGTCCTTTCGGCAAGCTTTTCCGCCCCGACAACTTCGTCTTCGGTCAGTCCGGTGCTGGTAACAACTGGGCCAAGGG

>56-1-Bt

CGGTGCTGCTTTCTGGCAGACCATCTCCGGCGAGCATGGTCTTGACAGCAACGGTGTCTACAATGGCACCTCTGAGCTCCAGCTCGAGCGCATGAGCGTTTACTTCAACGAGGTACGTTTGATGGGTACGCGTCGGAAGACGAGAGAGGACTCGATAAACTGACCCGCGTCAATAGGCTTCCGGTAACAAGTACGTTCCTCGCGCTGTTCTCGTCGATCTCGAGCCCGGTACCATGGACGCTGTCCGTGCTGGTCCCTTCGGTCAGCTTTTCCGCCCCGACAACTTCGTTTTCGGTCAATCCGGTGCTGGCAACAACTGGGCCAAGGGTCAC

  1. 4, authors should provide the morphology of fungal isolates with higher magnification power.

Response: Thanks for your good suggestion. First of all, we thank you very much for your excellent suggestion, we modified the figures, but we are not sure the figures that can meet the requirement. we are so sorry that we selected the magnification power of 40x,and these figures are the most clearest. The figures are following that:

  1. 6, unclear, please magnifies the image.

Response: Thanks for your good suggestion. We are so sorry for our carelessness. We have already checked the Figure 6 and modified them. They are following that:

  1. Line 415-417, please revise, isn’t clear, and there is a miswriting in the provided value (070). Also, please provide the obtained values for the effect of ozone on the disease development as percent to be clear for the readers. 

Response: Thank you very much for good question. we sincerely apologize for our mistake. We have already checked and modified, the sentence of “The development of postharvest disease was effectively suppressed in C. pilosula inoculated with 9 isolates after ozone fumigation treatment; and there was an ozone-exposure-time-dependent relationship with the inhibitory effect, for instance, the disease index in C. pilosula inoculated with Actinomucor elegans (7-1) after 1 and 2 h ozone exposure were 070, and 0.37 times higher than those in control (Figure 8A). The disease incidence in C. pilosula inoculated with F. acuminatum (14-1) after 1 and 2 h ozone exposure was 0.69, and 0.36 times higher than those in control (Figure 8B).” has been replaced with “The development of postharvest disease was effectively suppressed in C. pilosula inoculated with 9 isolates after ozone fumigation treatment; and there was an ozone-exposure-time-dependent relationship with the inhibitory effect. For instance, the disease indexes in C. pilosula inoculated with Actinomucor elegans (7-1) after 1 and 2 h ozone treatment were respectively 70%, and 37% higher than those in control (Figure 8A). The disease incidences in C. pilosula inoculated with F. acuminatum (14-1) after 1 and 2 h ozone exposure were respectively 69% and 36% higher than those in control (Figure 8B).” in the revised manuscript.

Reviewer 2 Report

This research work is an important contribution to the post-harvest control of C. pilosula disease a medicinal plants. It has a significant economic interest for C. pilosula producing countries like China. The analysis performed were satisfactory and well planned. The amount of samples, material and methods involved are consistent and relevant. In addition, this research also indicated the high occurrences of mycotoxins in C. pilosula and the proposed strategy of ozone treatment as effective strategies. This research is worthy of being published in this journal. However, some minor corrections need to be made to improve its overall value.

1. Authors should add reference for part 2.2. and give characteristic for ‘naturally occurring symptoms’. Line 96.

2. Authors should describe what they refer by typical ‘decay symptoms’  (Line- 102; same for line 153).

3. The treated fragments (Line 106) were inoculated onto potato dextrose agar (PDA) medium…. The term “Inoculated” is more convenient for microbe and authors should rewrite this expression

4. Authors should remove the inappropriate expression (Line 199)       …color colonies became complicated…

5. In Fig. 1 --- D has 7 individuals while the other images have 6.

6. In Section 2.6 and 2.7 the authors didn’t clearly describe what each treatment was made of.

7. Even more expert can know, the formula of index should be made clear for anyone to understand by defining “class frequency” or “score of rating class”

8. In the Line 180 the authors should delete “in the presence of” by “in”

9. 187 The experiment was done  which experiment the authors the refer to

10. Line 193, authors should rewrite this sentence to better start this paragraph and remove this expression “With the prolongation of storage time, it is”

11. C. pilosula was seriously diseased, (Line 202) authors should correct this expression.

12. Line 200, “…different color colonies became complicated” authors should correct or delete this expression.

13. Line 217, “On the 56th day of storage, 9 isolates were isolated”, The authors should replace “isolated” by “obtained” or other more convenient word.

14. Line  217 ……. 42nd may be  42 or 42th????

15. In section 3.3 part the authors should avoid repeating the information already in the table to increase concision. For example, the authors should add also column in the table the speed of growth on PD or the diameter of spore. Thus, in the text they should focus on presenting majors difference or interesting fact that they will discuss in the discussion part.

16. Molecular biological identification or 3.4 Molecular identification ?

17. In Figure 5, may be the letter B and C are missed the authors should correct it.

18. Figure 8 as each figure should stand alone, so the authors should add the code of each strain and they name of strain.

19. Line 478-501, please the authors should contract this part as it is not very related to the present work

20. Line 157, The above pathogens were verified according to Koch’s postulates….. the authors can discuss if the disease is caused by all or one pathogen or may be introduce the pathobiome concept that go beyond the postulate of Koch where no just one pathogen but many pathogens are interacting for disease. 

Author Response

Cover Letter

Dear Reviewer,

On behalf of our co-authors, we thank you very much for giving us this opportunity to revise our manuscript, we appreciate editor and reviewers very much for your positive and constructive comments and suggestions on our manuscript entitled “Isolation of Main Pathogens Causing Postharvest Disease in Fresh Codonopsis pilosula during Different Storage Stages and Ozone Control against Disease and Mycotoxin Accumulation”. (Manuscript ID: jof-2153731).

We have studied carefully reviewer’s comments and have made revision which marked up using the “Track Change” in the revised manuscript. We asked Dr. William, whose mother language is English, to edit the English grammar and sentence, and improve the English writing level. We also tried our best to revise our manuscript according to the comments. The revised manuscript was attached, and responses to their specific comments are detailed below.

We would like to express our great appreciation to you and reviewers for comments on our manuscript. Looking forward to hearing from you.

Thank you and best regards.

Sincerely yours,

Prof. Dr. Huali Xue

Gansu Agricultural University

Reviewer comments:

Reviewer: 2

  • Authors should add reference for part 2.2. and give characteristic for ‘naturally occurring symptoms’. Line 96.

Response: Thanks for your good suggestion. We have already checked, added the reference, and gave characteristic, the sentence of “Subsequently, the naturally occurring symptoms were observed and described.” has been replaced with “Subsequently, the naturally occurring symptoms were observed and described. Different pathogens cause different disease symptoms, for example, In the early stages of root rot, small brown spots appear on the surface of the lower fibrous or lateral roots, with mild decay. As the disease expands, it gradually spreads to the main roots. The roots gradually decay from the bottom upwards and become dark brown and waterlogged [9].” in the revised manuscript.

  • Authors should describe what they refer by typical ‘decay symptoms’ (Line- 102; same for line 153).

Response: Thanks for your good suggestion. We have already checked and modified.

Line-102 (now Line 384): “Pathogens were isolated and purified from C. pilosula based on typical decay symptoms [16].” has been replaced with “Pathogens were isolated and purified from C. pilosula based on typical decay symptoms (the appearance of mycelium and spores on the surface of C. pilosula) [16].”

Line-153 (now Line 432-436): “After an incubation period of 28 days, the disease symptoms were recorded [23].” has been replaced with “After an incubation period of 28 days, the disease symptoms were recorded, and different pathogens leaded to different disease symptoms [23].”

  • The treated fragments (Line 106) were inoculated onto potato dextrose agar (PDA) medium…. The term “Inoculated” is more convenient for microbe and authors should rewrite this expression.

Response: Thanks for your good comment. We have already checked and replaced, the sentence of “The treated fragments were inoculated onto potato dextrose agar (PDA) medium and cultured in darkness at 25 °C for 5 to 7 days.” has been replaced with “The treated fragments were placed onto potato dextrose agar (PDA) medium and cultured in darkness at 25 °C for 5 to 7 days.” in the revised manuscript.

  • Authors should remove the inappropriate expression (Line 199) …color colonies became complicated….

Response: Thanks for your good suggestion. We have already checked and replaced, the sentence of “When stored for 28 days, the different color colonies became complicated, some hyphae turned yellow, red, and green.” has been replaced with “When stored for 28 days, some yellow, red and green hyphae appeared on the surface of the C. pilosula.” in the revised manuscript.

  • In Fig. 1 --- D has 7 individuals while the other images have 6.

Response: Thanks for your carefulness. We are so sorry for our carelessness. We acknowledge that this mistake really should not make, but during the natural occurring symptoms stage we were only observing the occurrence of the disease in the C. pilosula and were not involved in the calculation of the disease incidence and disease index. Therefore, we thought that the number of C. pilosula had little or no effect on the experimental results. Again, we apologize for our carelessness.

  • In Section 2.6 and 2.7 the authors didn’t clearly describe what each treatment was made of.

Response: We are so sorry for our carelessness. We have added the sentence of “Each treatment contained three replicates, and one replicate included 50 samples.”

  • Even more expert can know, the formula of index should be made clear for anyone to understand by defining “class frequency” or “score of rating class”.

Response: Thanks for your good suggestion. We have added the defining “class frequency” or “score of rating class”. they are following that: “class frequency: the number of diseased plants at each rate; score of rating class: the diseased value for each rate”.

  • In the Line 180 the authors should delete “in the presence of” by “in”.

Response: Thanks for your good suggestion. We have already checked and replaced, the sentence of “5.0 g frozen sample was ground in the presence of liquid nitrogen, then was transferred to a 50 mL centrifuge tube with extraction solvent to extract mycotoxin.” has been replaced with “5.0 g frozen sample was ground in liquid nitrogen, then was transferred to a 50 mL centrifuge tube with extraction solvent to extract mycotoxin.” in the revised manuscript.

  • 187 The experiment was done which experiment the authors the refer to.

Response: Thanks for your good suggestion. The experiments mentioned in line 187 refer to the experiments done in parts 2.6 and 2.7. We have modified it as “The experiment of the effect of ozone on postharvest disease and mycotoxin accumulation of C. pilosula was done at least three times”.

  • Line 193, authors should rewrite this sentence to better start this paragraph and remove this expression “With the prolongation of storage time, it is”.

Response: Thanks for your good suggestion. We have already checked and replaced, the sentence of “With the prolongation of storage time, the disease development of freshly harvested C. pilosula became more severe (Figure 1).” has been replaced with “With the extension of storage time, the disease development of freshly harvested C. pilosula were more severe (Figure 1).

  • pilosula was seriously diseased, (Line 202) authors should correct this expression.

Response: Thanks for your good suggestion. We have already checked and replaced, the sentence of “When stored for 56 days, C. pilosula was seriously diseased, and the tissue became wrinkled, soft, and even rotten.” has been replaced with “After 56 days of storage, C. pilosula was seriously diseased, and the tissue were wrinkled, soft, and even rotten.

  • Line 200, “…different color colonies became complicated” authors should correct or delete this expression.

Response: Thanks for your good suggestion. We have already checked and replaced, the sentence of “When stored for 28 days, the different color colonies became complicated, some hyphae turned yellow, red, and green.” has been replaced with “When stored for 28 days, some yellow, red and green hyphae appeared on the surface of the C. pilosula.” in the revised manuscript.

  • Line 217, “On the 56th day of storage, 9 isolates were isolated”, The authors should replace “isolated” by “obtained” or other more convenient word.

Response: Thanks for your good suggestion. We have already checked and modified, the sentence of “On the 56th day of storage, 9 isolates were isolated, there was a new isolate named 56-1.” has been replaced with “On the 56th day of storage, 9 isolates were obtained, there was a new isolate named 56-1.

  • Line 217 ……. 42nd may be 42 or 42th????

Response: Thanks for your good suggestion. We are so sorry for our carelessness. We have already checked and replaced, the sentence of “There were no new isolates on the 42nd and 49th days.” has been replaced with “There were no new isolates on the 42th and 49th days.” in the revised manuscript.

  • In section 3.3 part the authors should avoid repeating the information already in the table to increase concision. For example, the authors should add also column in the table the speed of growth on PD or the diameter of spore. Thus, in the text they should focus on presenting major differences or interesting fact that they will discuss in the discussion part.

Response: Thank you for your excellent suggestion. We have modified them, deleted the description of margin, and added the growth speed of different isolates. They are following that:

Table 2. Morphological characteristics of pathogens isolated at different storage periods.

        Colony morphology

            Microscopic morphology

 Strain number

Front color

Back color

Texture

growth

speed(mm/d)

Conidium

Conidiophore

7-1

white

white

flocculent

24.57

spherical or

near spherical

sporangium

7-2

off-white

white

flocculent

21.5

spherical or

near spherical

sporangium

14-1

rose-red

rose-red

fluffiness

7.39

spindle-shaped

erect and branch

14-2

light pink

light pink

fluffiness

4.75

spindle-shaped

erect and branch

14-3

dark purple

dark purple

fluffiness

4.57

spindle-shaped

erect and branch

21-1

white

light yellow

fluffiness

4.29

spherical or

near spherical

erect and branch

28-1

grey green

white

Powdery orgrainy

8.71

spherical or

flat spherical

erect, broom

28-2

blue-green

tan

grainy

6.50

spherical or

flat spherical

pear-shaped or obovate

erect, broom

56-1

orange

orange

grainy

10.14

erect

  • Molecular biological identification or 3.4 Molecular identification?

Response: Thanks for your carefulness. We have already checked and replaced, the sentence of “3.4 Molecular biological identification of pathogens at different storage stages” has been replaced with “3.4 Molecular identification of pathogens at different storage stages” in the revised manuscript.

  • In Figure 5, may be the letter B and C are missed the authors should correct it.

Response: Thanks for your carefulness. We are so sorry for our carelessness. We have already checked the Figure 5 and added it. They are following that:

Figure 5. Gel electrophoresis images of PCR amplification products. A: ITS gel electrophoresis images; B: TEF gel electrophoresis images; C: TUB gel electrophoresis images.

  • Figure 8 as each figure should stand alone, so the authors should add the code of each strain and they name of strain.

Response: Thanks for your good suggestion. We have already checked the Figure 8 and modified it. They are following that:

Figure 8. The effect of ozone treatment on disease index (A) and disease incidence (B) of fresh C. pilosula infected by the 9 isolates at 56 days of storage. (7-1: Actinomucor elegans; 7-2: Mucor hiemalis; 14-1: Fusarium acuminatum; 14-2: Fusarium equiseti; 14-3: Fusarium oxysporum; 21-1: Clonostachys rosea; 28-1: Penicillium expansum; 28-2: Penicillium aurantiogriseum; 56-1: Trichothecium roseum) The different letters indicate significant difference during the same storage period (p < 0.05).

  • Line 478-501, please the authors should contract this part as it is not very related to the present work.

Response: Thanks for your good suggestion. We have already checked and revised, the paragraph of “At present, numerous studies have addressed the pathogens causing preharvest disease in Chinese herbs. Li et al. [28] isolated and identified the main pathogens causing Ligusticum chuanxiong root rot in Sichuan province of China, they were F. solani, F. oxysporum, Plectosphaerella cucumerina, and Phoma glomerata. Zhou et al. [29] isolated and identified the pathogens causing Astragalus root rot in the Moqi area of Inner Mongolia, and indicated that the predominant pathogens were F. oxysporum, F. solani, F. acuminatum, and F. equiseti. Wu et al. [30] verified that F. oxysporum, F. solani, and F. tricinctum were the main pathogens causing root rot of Coptis chinensis. Wang et al. [10] isolated and identified the main pathogen from the root rot of Panax ginseng as F. oxysporum. Chen et al. [31] isolated and identified the main pathogen from the sour rot of Pseudostellariae radix as F. oxysporum. The above reports suggested that Fusarium species are the dominant pathogens isolated from Chinese herbal medicines in various regions. However, the reported diseases caused by Fusarium spp. mainly address growth stages in the field (that is before harvest); there have been only a few reports regarding postharvest diseases of freshly harvested Chinese herbal medicines during storage. Of relevance, Chen et al. [32] isolated and identified P. crustosum, P. viridicatum, and P. aurantiogriseum from Angelica sinensis decoction pieces during storage; P. brevicompactum and P. aurantiogriseum were isolated from C. pilosula decoction pieces; and F. oxysporum and F. proliferatum were isolated from Licorice decoction pieces. Nevertheless, there are significant differences between Chinese medicinal decoction pieces and freshly harvested Chinese herbs as experimental materials. Chinese medicinal decoction pieces are mostly obtained from the Chinese herbal medicine market, these products have been already processed to be dry or dehydrated; alternately, freshly harvested Chinese herbs come from the field, without any processing and treatments after harvest such as drying or dehydration. Therefore, fresh C. pilosula is prone to mouldiness and decay, owing to the higher water content and the abundance of nourishing substances that allow the growth of pathogens.” has been replaced with “At present, numerous studies have addressed the pathogens causing preharvest disease in Chinese herbs, these reports indicated that Fusarium species are the dominant pathogens to cause preharvest disease in the various regions [28-31]. However, there are limited reports on postharvest diseases of freshly harvested Chinese herbal medicines during storage. Of relevance, Chen et al. [32] isolated and identified P. crustosum, P. viridicatum, P. aurantiogriseum and P. brevicompactum from Angelica sinensis and C. pilosula decoction pieces during storage. Nevertheless, there are significant differences between Chinese medicinal decoction pieces and freshly harvested Chinese herbs as experimental materials. Chinese medicinal decoction pieces are mostly obtained from the Chinese herbal medicine market, and these products are available in dry or dehydrated forms. On the other hand, freshly harvested Chinese herbs come from the field, without any processing and treatments after harvest such as drying or dehydration. Therefore, fresh C. pilosula is more susceptible to molds and decay, owing to the higher water content and the abundance of nourishing substances that allow the growth of pathogens.” in the revised manuscript.

  • Line 157, The above pathogens were verified according to Koch’s postulates….. the authors can discuss if the disease is caused by all or one pathogen or may be introduce the pathobiome concept that go beyond the postulate of Koch where no just one pathogen but many pathogens are interacting for disease.

Response: Thank you for your good suggestion. Koch’s postulate is to verify the pathogenicity of isolate. Indeed, in the experiment, we observed the initial disease symptom that were caused by several pathogens, these pathogens were interacting for the postharvest disease. Then we isolated and identified the pathogens during different storage, and respectively inoculated with these obtained pathogens to the healthy C. pilosula. The disease symptom we observed was caused by one pathogen. Therefore, we think it went beyond Koch’s postulate. We added this part of “All the 9 isolates could cause different postharvest disease, and the pathogens were verified according to Koch’s postulates. However, in fact, the results went beyond Koch’s postulate. Because Koch’s postulate is mainly applied to one pathogen. However, we initially observed the postharvest disease that many pathogens were interacting for disease, and we isolated and identified the different pathogens, then inoculated the healthy C. pilosula with the obtained pathogen, finally observed the different disease symptom caused by different pathogen. Therefore, we thought Koch’s postulate had limitation for the results in this study.” in the revised manuscript.
